# Reconstructing Six Decades of Surface Temperatures at a Shallow Lake

**Xin Zhang [1,2], Kaicun Wang [1,*]**  **, Marieke A. Frassl [3] and Bertram Boehrer [2]**

[1]   State Key Laboratory of Earth Surface Processes and Resource Ecology, College of Global and Earth System Science, Beijing Normal University, Beijing 100875, China; xzhang0828@mail.bnu.edu.cn

[2]   Department of Lake Research, Helmholtz Centre for Environmental Research, 39114 Magdeburg, Germany; bertram.boehrer@ufz.de

[3]   Australian Rivers Institute, Griffith University, QLD 4111 Nathan, Australia; marieke.frassl@gmail.com

*   Correspondence: kcwang@bnu.edu.cn

**Abstract:** Lake surface water temperature (LSWT) plays a fundamental role in the lake energy budget. However, direct observations of LSWT require considerable effort for acquisition and hence are rare relative to a large number of lakes. In lakes where LSWT has not been covered sufficiently by in situ measurements, remote sensing and lake modeling can be used to produce a fine spatio-temporal record of LSWTs. In our study, the Moderate-Resolution Imaging Spectroradiometer (MODIS) LSWT was used to compare with in situ data at the overpass times over the six sites in Lake Chaohu, a large shallow lake in China. MODIS-derived LSWT reflected the variation of lake surface temperature well, with a correlation coefficient of 0.96 and a cool bias of 1.25 °C. The bias was modified by an "Upper Envelop" smoothing method and then employed to evaluate the general lake model (GLM) performance, a one-dimensional hydrodynamic model. The GLM simulations showed good performance compared with MODIS LSWT data at an interannual time scale. A 57-year record of simulated LSWT was hindcast by the well-calibrated GLM for Lake Chaohu. The results showed that LSWT decreased by 0.08 °C/year from 1960 to 1981 and then increased by 0.05 °C/year. These trends were most likely caused by a cooling effect of decreased surface incident solar radiation and a warming effect of reduced wind speed. Our study promoted the use of MODIS-derived LSWT as an alternative data source, and then combined with a numerical model for inland water surface temperature, and also further provided an understanding of climate warming effect on such a shallow eutrophic lake. **Key points:** (1) Moderate-Resolution Imaging Spectroradiometer (MODIS) lake water surface temperature (LSWT) was validated with real-time in situ data collected at Lake Chaohu with high accuracy; (2) MODIS LSWT was modified by the bias correction and employed to evaluate a one-dimensional lake model at interannual and intraannual scale; The LSWT hindcast by a well-calibrated model at Lake Chaohu decreased by 0.08 °C/year from 1960 to 1981 and increased by 0.05 °C/year from 1982 to 2016.

**Keywords:** lake water surface temperature; MODIS; lake modeling; climate change

## 1. Introduction

Lake surface water temperature (LSWT) is a fundamental driver of lake ecosystem structure and its energy budget [1]. LSWT affects the stability and the timing of stratification [2–5], the rate of lake evaporation [6], the gas exchange [7], mixing processes [8], and duration of ice coverage in some temperate regions [9]. Responding to climate changes, it has been reported that summer LSWTs have increased by 0.34 °C per decade [10,11], even faster than the rate of local air temperature [12–14].

Such a warming trend has also been documented in the Laurentian Great Lakes regions [15] and Europe [16–18].

Compared to the number of lakes, long-term records of lake surface temperature measurement from thermistors are scarce and largely geographically restricted to North America and Europe [12,19,20], as the consistent and high-frequency in situ measurements of LSWT require considerable effort and human power to maintain technical equipment. In China, or developing countries, only a handful of lakes out of thousands have long-term and high-frequency LSWT measurements. As a consequence, long-term variability of lake temperature and responses to climate change are difficult to retrieve. In conclusion, an approach to extend time series into the past is highly desirable.

Remote sensing appears to be an attractive way to complement traditional measurement with a fine spatio-temporal resolution. Several regional-scale LSWT products are also available with the acceptable accuracy, e.g., LSWT for European Alpine lakes (1989–2013) using the Advanced Very High Resolution Radiometer (AVHRR) [21,22]; LSWT for Tibetan Plateau lakes using the Moderate-Resolution Imaging Spectroradiometer (MODIS) land surface Temperature products [23] and AVHRR products [24], Arclake–LSWT products from 1995–2012 using the Along Track Scanning Radiometer (ATSR) series [25]. Those satellite products have been widely used to investigate LSWT variation in global and regional scale [16,26]. Previous studies on MODIS temperature products have demonstrated reasonable accuracy on the land surface [27–29]. It is suggested that the accuracy is even better on the water surface due to its large homogeneity compared to the land surface [29].

However, the temporal extension of LSWT record with satellite data is far from a simple implementation. Firstly, the daily LSWT data from AVHRR or MODIS still suffer from gaps due to cloud-contaminated pixels. For example, the MODIS-derived LSWT dataset in Tibetan Plateau lakes is contaminated by cloud cover, particularly for the nighttime in summer [23]. Several approaches were adopted to reconstruct data over the gaps to build a reliable daily time series of LSWT, such as a harmonic analysis of time series (HANTS) with fitted values from sinusoidal function [30], Percentile Filter and Lowess Filter [23]. Also, satellite-derived LSWT and in situ measurement represent the water temperature of spatially and temporally different scales. While in situ measurements referred to a certain depth at one location (referred to bulk temperature), satellite data record the temperature of the upper skin layer by averaging over a larger spatial area (referred to skin temperature). The difference between the skin temperature from satellite and the bulk temperature depends on wind speed, atmospheric aerosols, and surface elevation in a complex manner [19,31–34]. For example, a noticeable difference between the bulk and skin temperature was found in Lake Tahoe [35]. Donlon, et al. [33] found the difference was obvious especially when at low wind speeds and high radiation in the morning. The relationship between skin and bulk temperature was complicated. An enhanced cool bias was observed when skin surface temperature was larger than 28 °C [36]. Given that, it was worth to make a comparison by utilizing in situ data with high frequency matching the exact acquisition time of the satellite products. It could assess the accuracy of MODIS retrieved LSWT data and make bias correction for application in regions where records of LSWT observation were limited.

As an alternative, lake models also can be used to reconstruct long term LSWT [34,37–40]. One-dimensional (1-D) physical-based models have been widely used in recent years. Although they do not resolve lake temperature simulation horizontally (as three-dimensional models do), 1-D dimensional models can offer a good compromise between computational efficiency and physical reality with minimal calibration requirements and fewer requirements of input data compared with two- and three-dimensional models [41–45]. The abilities of one-dimensional models have been documented well to reproduce temperature profiles, mixing regimes, and reflect vertical effects on surface temperature variation [40,46,47]. The evaluation on a wide range of lakes in various climate zones, such as in Lake Geneva [44], Lake Kossenblatter [48] in mid-latitude zones, Lake Valkea-Kotinen [49] in a boreal climate, and Lake Kivu [43] in a tropical climate, showed that the dynamics of lake surface temperature can be accurately captured in a one-dimensional model.

In this study, we highlighted a direct overpass comparison between skin temperature measurements obtained from satellites and bulk temperatures of water column layers from observations. This comparison would enable the relationship between skin temperature and bulk temperature. We further proposed a novel strategy for adjusting remotely sensed skin temperature to the bulk temperature of surface layer, then used the data for calibration and validation in a 1-D hydrodynamic lake model. The scientific merit of this work is to provide an efficient bias correction technique for extrapolating remotely sensed skin temperature observation and produce an alternative data source for the lake model. Another scientific merit is that the well-calibrated lake model can forecast lake surface temperature with reasonable accuracy using key meteorological variables, such as wind speed and solar radiation. This is important because it could give a further understanding of how climate change affected a shallow and large lake, especially in an eutrophic condition. In our study, we selected a large shallow lake, Lake Chaohu, located at mid-latitudes in the east of China. The high-frequency in situ LSWT observation, MODIS temperature products, and a general lake model (GLM) were implemented to produce comparable long-term LSWT time series. We addressed the objectives in following steps: 1) evaluate the reliability of MODIS derived LSWT by high-frequency in situ water temperature observations; 2) assess the performance of the GLM simulation by in situ LSWT and MODIS LSWT at different time scales; 3) reproduce LSWT over a period of 57 years and quantify climatic variability of LSWT and its response to a changing climate.

## 2. Data and Methods

### 2.1. Study Area

Lake Chaohu (31°43′–31°72′ N, 117°29′–117°85′ E) is a polymictic lake in Anhui Province in the lower reach of the Yangtze River (Figure 1). A large portion of the inflow is from Nanfei River, Hangbu River and Yuxi River and one outflow connecting with Yangtze River. It is the fifth largest freshwater lake in China. The mean depth of the lake is 3 m (maximum depth is 7 m), and the surface area is about 780 km$^2$. The nearest weather station is named Hefei (31°53′ N, 117°15′ E). The lake basin experiences a northern subtropical monsoon climate with an average annual rainfall of 1100 mm. The annual average air temperature is 15–16 °C.

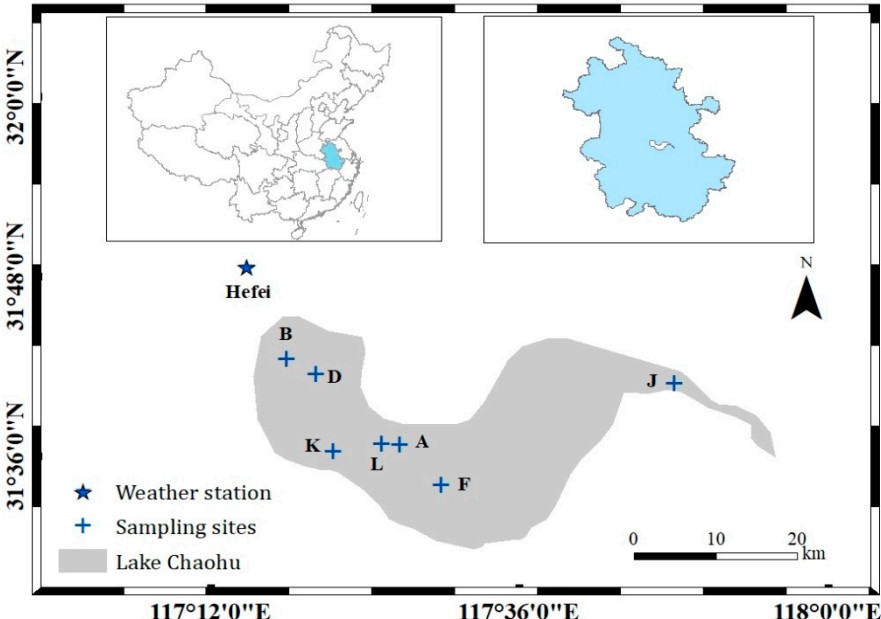

**Figure 1.** Location of Lake Chaohu at the Anhui Province in China, sampling sites for water temperature, and the weather station named Hefei.

## 2.2. In Situ Lake Temperature Data

In situ temperatures in Lake Chaohu were measured at intervals of 15 min with loggers in 6 locations distributed over the lake at a depth of 0.3 m below the surface (Figure 1, HOBO Onset, TidbiT; accuracy ±0.2 °C) [50]. The sites were distributed at the lake center, the bay center and bay mouth (Figure 1). These in situ temperatures were acquired from 1 January 2016 to 31 December 2016 with a data gap from June to August due to malfunctioning. The daily average temperature was calculated at each site and then averaged over all the sites for the entire lake. Also, monthly observations were obtained from 2008 to 2016 from Kong, et al. [51], at a depth of 0.2~0.5 m near the lake center usually at 10:00.

## 2.3. Moderate-Resolution Imaging Spectroradiometer (MODIS) Lake Surface Temperature

### 2.3.1. Pre-Processing

Daily LSWTs were obtained from MODIS land surface temperature products (MOD11A1, MYD11A1, version 6), available on National Aeronautics and Space Administration (NASA) Earth Observation System Data and Information System [52]. The spatial resolution was 1 km and the split-window algorithm was used to retrieve surface temperature from band 31 and band 32 in products and cloud-contaminated temperatures have been removed from these products. The "land" surface temperature products estimated by satellite instruments include both land and inland water [29]. Because of diurnal variation of lake surface temperature [53], LSWTs were obtained from four overpasses per day: Terra overpasses at local times around 10:30 and 22:30, and Aqua overpasses at local times around 1:30 and 13:30. Overpass times could vary slightly and could be extracted from MODIS data files. MOD11A1 data (Terra) were available since 2001 and MYD11A1 data (Aqua) since 2002. In this study, we used data from MOD11A1 and MYD11A1 from 2002 to 2016.

The temperature map for the whole lake from MODIS temperature products was extracted by a lake shapefile mask from high-resolution data of Landsat Thematic Mappez (TM) image at 30 m with a 3-pixels buffer zone along the shore to avoid errors due to fluctuations of land–water interfaces or the variance of year-to-year lake boundary [23]. The temperature was valid only if the ratio of acceptable quality pixels to water pixels was larger than 50% in the whole mask. The pixels were used through the strict quality control according to quality control flag and quality assessment information provided by MOD11A1 and MYD11A1. The reliable quality pixels were accepted flagged with "good quality", "average LST error <= 1K", "average emissivity error <= 0.01", and "average emissivity error <= 0.02". The pixels with the other flags were rejected.

### 2.3.2. Comparison with In Situ Data

In our study, we compared MODIS-derived LSWT and in situ observation at the site scale and over the whole lake, respectively. The real-time comparisons were evaluated between MODIS-derived LSWT and in situ water surface temperature at six sites across Lake Chaohu. The site J was not included because it was located too close to the lake boundary. The 3 × 3 pixels matrix centered on sample sites was extracted from MODIS LSWT images. The area of this pixel matrix was large enough to represent the temperature estimated from MODIS images [26]. The exact overpass time information of MODIS LSWT was extracted and then the corresponding time of observed LSWT was selected from 15-min high-frequency in situ data. The mean temperature of the matrix was calculated. We further computed MODIS LSWT values over the whole area of the lake and then averaged all four records per day, which was identified as the daily temperature of LSWT from MODIS. The outliers of MODIS-derived LSWT were examined and rejected when the values deviated from climatological temperature by more than 3 times the variation. The climatological temperature was calculated by multi-year average temperature from MODIS.

### 2.3.3. Post-Processing

Although the cloud-contaminated pixels have been removed by the cloud mask [29], abnormally low values of LSWT were observed owing to the undetected cloud pixels which were much colder than lake surface temperature [29]. Also, systematic biases were found in MODIS-derived LSWT, such as cool bias from skin-bulk temperature difference, instrument noise, drift, sun glint, and split window coefficients for inland water [19,31,54,55]. To reduce the difference above and obtain a reliable daily time series LSWT, we adapted a novel method named "Upper Envelop" smoothing to minimize the difference. The upper envelop smoothing method adopted a simple linear interpolation method to primarily fill data gaps, caused by bad-quality pixels, and then adopted the three-point upper envelop smoothing method by Equation (1), according to Gu, et al. [56], to more effectively obtain the "best estimated" LWST from MODIS products. It was implemented to the solve negative noise reduction problem [56,57].

$$LST_{u-e}(t) = \max[LST(t), (0.5 \times LST(t) + 0.25(LST(t-1) + LST(t+1)))] \qquad (1)$$

where $LST_{u-e}(t)$ represented MODIS temperature from Upper envelop method, $LST(t)$, $LST(t-1)$ and $LST(t+1)$ represented the data on the $t$ th day, the previous time-step and the following time-step.

### 2.4. Quantitative Evaluation

The quality of LSWT from different data sources was quantified by an evaluation matrix including bias, standard error (STD), and root mean square error (RMSE). Bias was the average difference between MODIS-derived LSWT model simulation and observation, which was expected to evaluate under- or over-estimation. STD was the standard deviation of the difference, which was expected to evaluate the heterogeneity or variance of two datasets. RMSE quantified the overall accuracy of estimated LSWTs, as it combined the notions of bias and standard error. The LSWT and the related atmospheric variables were also analyzed by linear trend with robust analysis, and Pearson correlation coefficient.

The change-point test of LSWT time series was detected by two methods which have been widely used in many studies [58–61]: the sequential t-test of regime shift [62] and a non-parametric and robust against outliers method, called CUSUM (the cumulative sum) [63], combined with the 1000-times bootstrap analysis to determine the confidence level.

## 3. The General Lake Model (GLM)

### 3.1. Introduction of the General Lake Model

The general lake model (GLM, [64]), was a one-dimensional hydrodynamic model intrinsically assuming horizontal homogeneity. The model adopted a vertical Lagrangian layer structure and each layer had a unique density computed from the simulated salinity and temperature. The layer thickness was flexible, i.e., layers follow the vertical water movement to minimize numerical diffusion. Layers could split when they grow too voluminous or combine when sufficient energy became available to overcome density stratification between adjacent layers. GLM was particularly suited for long-term investigation, and required only little site-specific calibration [41]. The GLM model allowed users to simulate vertical profiles of temperature, salinity, and density. Mixing and surface layer dynamics were modeled at the confluence of adjacent layers and were dependent on a turbulent kinetic energy budget and potential energy required for mixing. Recent studies showed that GLM has been tested for water temperature in globally distributed lakes of 104 km$^2$ to 579,000 km$^2$ surface area [41] and more than 10,000 lakes in the United States [65].

### 3.2. Input Data

The input meteorological variables to force the GLM model (v2.4.0) were wind speed, air temperature, relative humidity, precipitation, shortwave radiation, and longwave radiation. We chose

measurements of wind speed, air temperature, relative humidity, precipitation and sunshine duration from the weather station named Hefei, shown in Figure 1, provided from the China Meteorological Administration dataset during the study period of 1960–2016. The dataset was applied to the strict quality control and of good quality [66]. The climatological information of six variables could also be found in Figure 2.

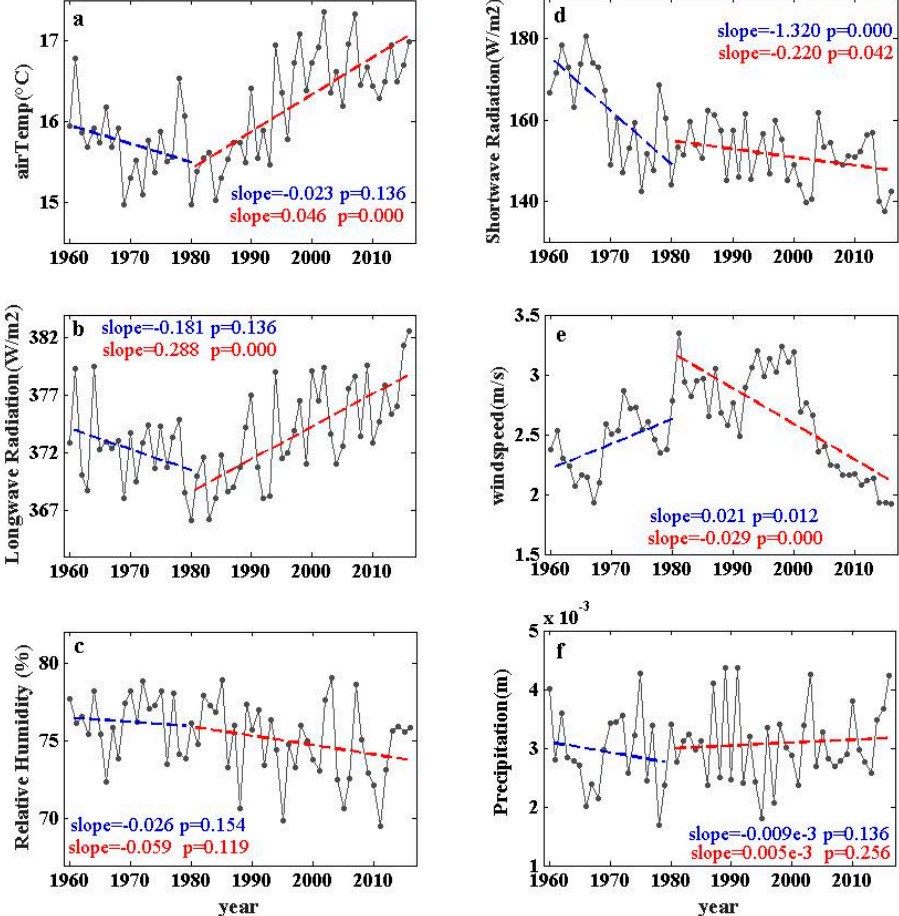

**Figure 2.** The annual value of atmospheric variables, air temperature (**a**), longwave radiation (**b**), relative humidity(**c**), shortwave radiation (**d**), wind speed (**e**), and precipitation (**f**) from 1960 to 2016. The linear trend for each variable is given during the period of 1960–1980 (blue dashed line) and the period of 1981–2016 (red dashed line).

The incoming shortwave solar radiation (referred to as shortwave radiation in the following context) was calculated by observed sunshine duration according to a method by Yang, et al. [67], which was subsequently improved by Wang [68]. It has been proved that shortwave radiation derived from sunshine duration reflects the effects of clouds and aerosols on shortwave radiation [68,69], and performed well at a regional and global scale [70–72]. Calculated shortwave radiation showed good accuracy when compared with shortwave radiation observation. Due to longer availability, the derived shortwave solar radiation revealed a dimming before the 1980s and a brightening afterward [69,70,73]. The sunshine duration derived shortwave radiation agreed well with satellite retrievals, reanalysis, and climate model [69]. Furthermore, the sunshine duration derived shortwave radiation corrected the inhomogeneity of observed shortwave radiation caused by the sensitive drift and instrument replacement and could reflect variability from diurnal to decadal time scales [69].

The incoming longwave solar radiation (referred to as longwave radiation) was estimated following the method from Brutsaert [74] to determine the clear-sky radiation and then corrected to all-sky

radiation by applying the cloud-sky equation from Crawford and Duchon [75]. This combination was adopted by the lake model internally and commonly used to estimate the longwave radiation [76–79].

### 3.3. Model Simulation

The model simulation required lake basic characteristics, input meteorological data, initial conditions, and a series of parameters. The model started on the first day of simulation period, such as 1 January 2016, and then ran forced by input meteorological data, producing daily water surface temperature. In our simulation, the heat effects of inflows and outflows were neglected, as inflow and outflow temperature differences were less than 1 °C according to previous observations (see Table 1 in Zhang, et al. [80] and Huang, et al. [81]), and residence times (~180 days [50]) were too long for a visible effect on water temperature. Also, the previous study showed the inflows and outflows in Lake Chaohu had a limited effect on the thermodynamics and currents in the lake, while energy exchange with atmosphere on lake surface had a more evident impact [82]. Hence, the effect of in- and outflows on lake surface temperature was not considered in our simulation.

**Table 1.** Statistic information about the comparison between Moderate-Resolution Imaging Spectroradiometer (MODIS)-derived lake surface water temperature (LSWT) and observed LSWT from six sites in 2016. The pairs were count for the comparison, and the bias, standard error (STD), R square, and root mean square error (RMSE) were calculated.

| Sample Site | Acquisition Time | Pairs | $R^2$ | Bias | STD | RMSE |
|---|---|---|---|---|---|---|
| Site A | Terra Day | 64 | 0.98 | −0.82 | 1.41 | 1.62 |
| | Terra Night | 67 | 0.99 | −1.94 | 1.19 | 2.27 |
| | Aqua Day | 44 | 0.98 | −0.50 | 1.28 | 1.36 |
| | Aqua Night | 58 | 0.98 | −1.78 | 1.27 | 2.18 |
| Site B | Terra Day | 50 | 0.97 | −0.71 | 1.50 | 1.64 |
| | Terra Night | 53 | 0.98 | −1.97 | 1.16 | 2.28 |
| | Aqua Day | 28 | 0.97 | −0.60 | 1.79 | 1.85 |
| | Aqua Night | 39 | 0.98 | −1.89 | 1.21 | 2.23 |
| Site D | Terra Day | 51 | 0.98 | −0.67 | 1.35 | 1.50 |
| | Terra Night | 49 | 0.98 | −2.12 | 1.22 | 2.44 |
| | Aqua Day | 31 | 0.95 | −0.63 | 1.99 | 2.06 |
| | Aqua Night | 40 | 0.98 | −1.88 | 1.24 | 2.25 |
| Site F | Terra Day | 37 | 0.90 | −0.95 | 1.87 | 2.07 |
| | Terra Night | 38 | 0.96 | −1.82 | 1.16 | 2.15 |
| | Aqua Day | 23 | 0.90 | −0.53 | 2.04 | 2.06 |
| | Aqua Night | 30 | 0.96 | −1.71 | 1.16 | 2.06 |
| Site K | Terra Day | 11 | 0.80 | −0.72 | 1.80 | 1.86 |
| | Terra Night | 9 | 0.85 | −1.22 | 1.53 | 1.89 |
| | Aqua Day | 4 | 1.00 | na | 0.54 | 1.09 |
| | Aqua Night | 6 | 0.95 | −1.99 | 1.25 | 2.29 |
| Site L | Terra Day | 47 | 0.96 | −0.71 | 1.86 | 1.97 |
| | Terra Night | 52 | 0.98 | −1.68 | 1.31 | 2.13 |
| | Aqua Day | 29 | 0.95 | −0.25 | 2.34 | 2.31 |
| | Aqua Night | 43 | 0.99 | −1.69 | 1.27 | 2.10 |

For most parameter settings, we used default values in our simulation, except for the light extinction coefficient and wind scaling factor. The Secchi depth (SD) was commonly around 0.3 m in spring and 0.48 in late autumn [83,84]. Hence a light extinction coefficient (named "$K_w$" in GLM) in the range of 3–6 m$^{-1}$ was appropriate, based on the relationship of $K_w$ and Secchi Depth from Liu, et al. [85]. The wind speed records were from the weather station at a considerable distance from the lake, and hence a wind scaling factor of 0.6–0.9 was applied based on our knowledge about the uncertainty of parameters and appropriate limits from previous applications of GLM ([65,86]. In

our study, the best-fit parameters were calibrated and adjusted to minimize the RMSE based on the comparison of simulated data with the observed data in 2016. The observed data referred to both daily time series of MODIS-derived LSWT and in situ data. The validation and evaluation of the GLM model with best-fit parameters were performed from 2002 to 2016 by the statistical criteria RMSE, bias, STD.

## 4. Results

### 4.1. Evaluation of MODIS Lake Surface Water Temperature (LSWT) with 15-Minute In Situ Data

MODIS LSWT was compared with in situ data at six sites across Lake Chaohu in 2016. The exact overpass time information of MODIS LSWT was extracted to make a comparison with high frequency observed LSWT in 2016. The comparison of MODIS LSWT and in situ data showed a good agreement (Figure 3) with the average correlation coefficient of 0.96 at six sites, and with the bias of 1.25 °C, and the STD of 1.45 °C, and the RMSE of 1.98 °C, respectively (Table 1).

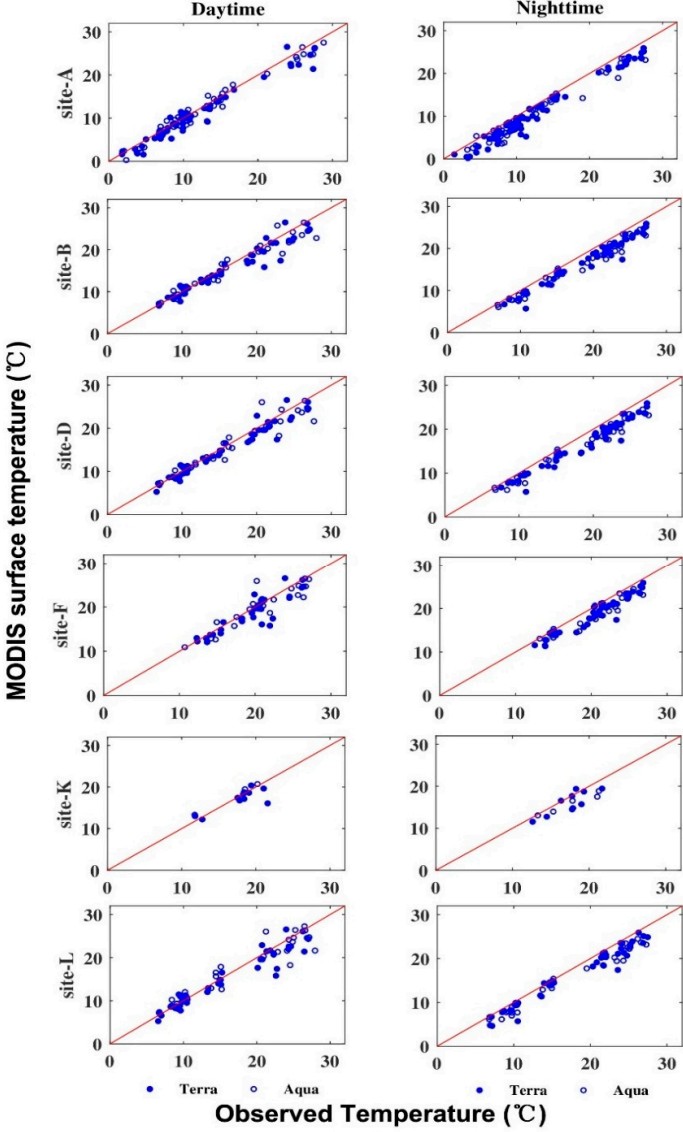

**Figure 3.** Scatterplots of comparisons of LSWT from MODIS and in situ data at six sites across Lake Chaohu in 2016. The left panel is the daytime comparison (~10:30 am from Terra and ~1:30 pm from Aqua), and the right panel is the nighttime comparison (~10:30 pm from Terra and ~1:30 am from Aqua).

The correlation was at nighttime than at daytime better between MODIS–derived LSWT and observation (see Figure 3 and Table 1). The STD amounted to 1.16 °C to 1.53 °C at nighttime and 1.35 °C to 2.3 4 °C at daytime. This could be attributed to better spatial homogeneity of lakes surface temperature during nighttime, and the absence of solar heating during nighttime [29]. In addition, the cool bias and STD of LSWT were smallest in Aqua daytime, as the overpass time for Aqua daytime was around 13:30, and lay close to the peaks of the diurnal LSWT cycle.

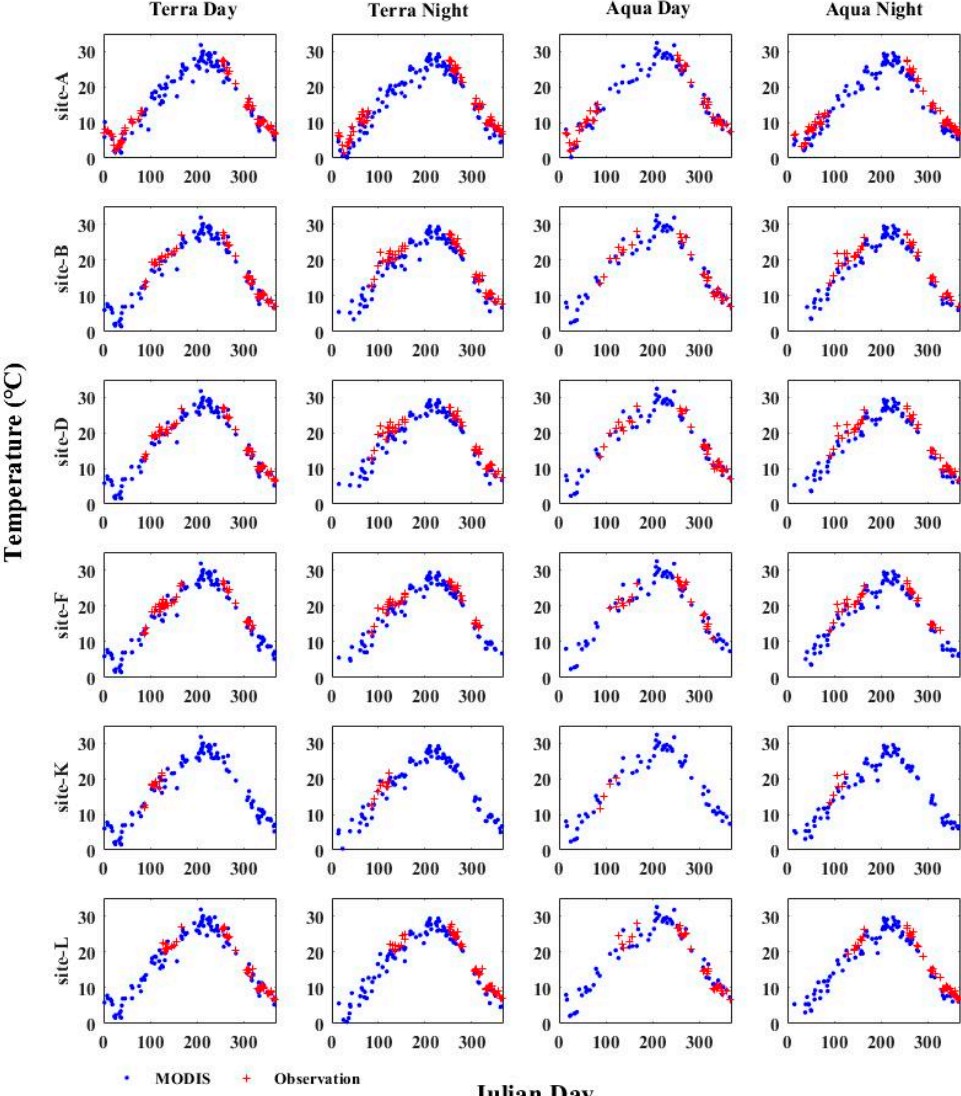

**Figure 4.** Scatterplots of annual variation of MODIS LSWT products and observed LSWT at six sites across Lake Chaohu in 2016. The first and second vertical panels represented at ~10:30 am and at ~10:30 pm. The third and fourth vertical panels represented at ~1:30 pm and ~1:30 am.

As shown in Figure 4, a cool bias of MODIS was detected across all six sites when real-time match-ups were compared. The bias varied from −0.50 °C to −0.99 °C at daytime and from −1.22 °C to −2.12 °C at nighttime. The larger bias at nighttime was likely owing to temperature gradients in the shallow lake that the upper layer of water was stratified during the day and mixed well during the night [4]. Due to the high turbidity, Lake Chaohu absorbed most of incoming shortwave radiation used for heating water in the upper layers. During the daytime, the water temperature in the upper layers was quite mixed, but showed a larger gradient between a thin surface and a sub-surface layer. While at nighttime, the mixing was less complete and water temperature was stratified without heating from

radiation, and hence the bias was larger between skin and bulk temperature. A similar situation has been documented in Lake Taihu, another shallow and turbid large lake further downstream in the Yangtze catchment [87].

Based on the observation in six sample sites, the highest average temperature was 17.34 °C and the lowest value was 16.92 °C. Also, the bias varied from −0.50 °C to −0.99 °C at daytime and from −1.22 °C to −2.12 °C at nighttime between observation and MODIS data among six sample sites. Hence, the heterogeneous temperature was small on the surface layer. Then we retained 280 matching pairs to validate the MODIS-derived LSWT with in situ observations over the whole lake area in 2016. The results showed a significant correlation with the $R^2$ value of 0.98 (Figure 5). There was also a noticeable cool bias of −1.76 °C when daily time series of MODIS-derived LSWT were compared with in situ measurements.

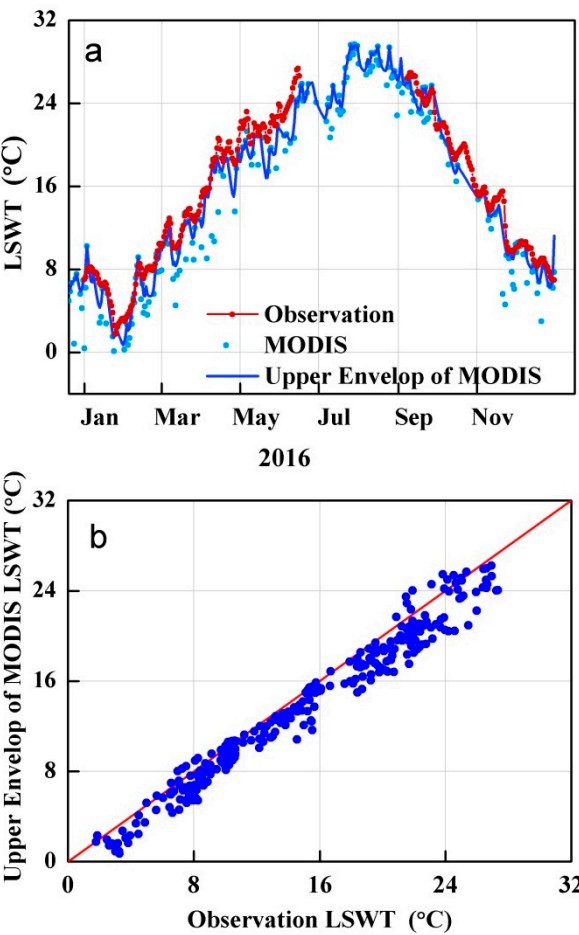

**Figure 5.** Comparison of daily LSWT between MODIS and in situ measurement over the whole lake area in 2016. Panel (**a**): the daily time series of LSWT from MODIS temperature product (symbols), and upper envelop smoothing correction (blue line), and in situ observation (red dot-line); panel (**b**): comparison of observations with upper envelop of MODIS-derived temperature.

When MODIS-derived LSWT was processed by the upper envelop method, and the cool bias reduced to 0.98 °C, as shown in Figure 5. The three-point "upper envelope" method worked well as all single measurements were either on the envelope or below it. The results revealed most of the temperature dynamics seen in in situ measurements qualitatively and quantitatively, provided enough support for the method.

The monthly LSWT observations from 2008 to 2016 were also used to validate the performance of MODIS and MODIS–UE (Upper Envelop line) LSWT, especially to bridge the data gap during summer (from June to August) due to malfunctioning of the high-frequency loggers. Monthly observations

were obtained from Kong, et al. [51], sampled monthly at a depth of 0.2–0.5 m near the lake center usually at 10:00. As shown from Figure 6, the seasonal cycle of LSWT fitted well between MODIS derive LSWT and observation, with the correlation coefficient of 0.98 and the bias of −0.35 °C. The bias was −2.05 °C in summer, higher than other spring (−1.09 °C), autumn (−0.77 °C) and winter (−0.55 °C).

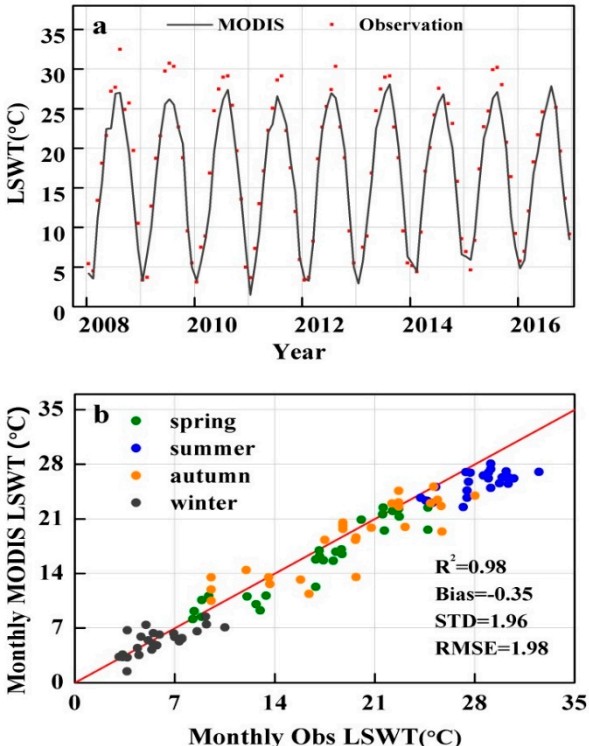

**Figure 6.** Comparison of monthly LSWT between MODIS–UE (Upper Envelop line) and Observation from 2008 to 2016. Panel (**a**): the red symbols are monthly observation, and the black line is the MODIS UE-derived LSWT. Panel (**b**): the scatterplot of the comparison between MODIS–UE and monthly observation LSWTs. The green dots represent the monthly value in spring (March, April, and May). The blue dots represent in summer (June, July, and August). The orange dots represent in autumn (September, October, and December). The black dots represent in winter (December, January, and February). The red solid line is 1:1 line.

It seemed that the LSWT from MODIS–UE data underestimated the surface summer during the summertime, especially when the temperature was above 27 °C. During summer, the monthly observation of LSWT was higher by 1.89 °C than MODIS-derived LSWT from 2008 to 2016. From June to September, the peak of shortwave radiation and lower wind speed was found in Lake Chaohu. The surface layers of lake would heat more and the skin effect through cooling would be more pronounced. In addition, the wind speed was low, and the skin layer could persist as it was not destroyed by wind mixing. Generally, a time series of upper envelope over the MODIS data (referred to as MODIS–UE LSWT), filled gaps properly and reduced uncertainty. The upper envelop curve of MODIS showed a good performance generally over the whole period, whereas an obvious cool bias remained in the summertime.

*4.2. The GLM Model Evaluation*

LSWT simulations compared well with MODIS–UE LSWT in 2016 (Figure 7). Bias and STD were 1.8 °C and 1.92 °C, and the correlation coefficient was 0.98. The variations were similar between MODIS–UE LSWT and GLM simulation. Then, LSWT simulation was also compared with MODIS–UE LSWT when excluding the data in the summertime, with the bias and STD value of 1.09 °C and 1.53 °C, respectively (as shown in Figure 7). The large bias of LSWT simulation was possibly contributed to

the data in the summertime, as a cool bias was found clearly in the summertime. As the observation was missing during the summer of 2016, it was difficult to validate whether MODIS–UE LSWTs or simulations were corrected during the summer, especially the cool bias might have been larger than in other seasons due to the peak value of solar radiation and low wind speeds [54,88]. Hence, the monthly observations from 2008 to 2016 and the MODIS LSWT from 2002 to 2016 were also employed to validate the GLM simulations performance.

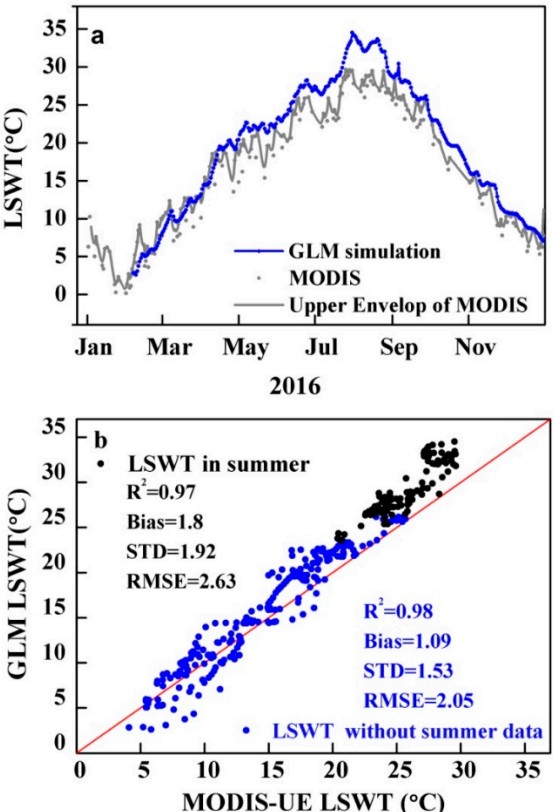

**Figure 7.** Comparison of daily LSWT between MODIS, MODIS–UE and GLM simulation in 2016 in Lake Chaohu. Panel (**a**): the blue line is the GLM simulation with best fit parameter (the minimum RMSE), and the grey line is the upper envelop of MODIS LSWT, which is from MODIS-derived LSWT (grey symbols); panel (**b**): the black dots represent the comparison of MODIS–UE temperature and the GLM simulation in summer, while the blue dots represent the comparison without summertime data in 2016.

The validation of the GLM simulations was accomplished by comparison with monthly observed LSWT from 2008 to 2016 and with MODIS LSWT from 2002 to 2016 (Figures 8–10). Simulation agreed well with MODIS–UE LSWT (Figure 8). The seasonal cycle was reflected both in MODIS–UE LSWT and simulations, but the LSWT from MODIS showed a large deviation to lower values in summer when compared with GLM simulations. As the data comparisons were shown at the monthly scale (Figure 9), the correlation coefficient was 0.96 and the bias amounted to 0.66 °C. The correlation coefficient was 0.94 and the bias was 0.25 °C when summer values were excluded.

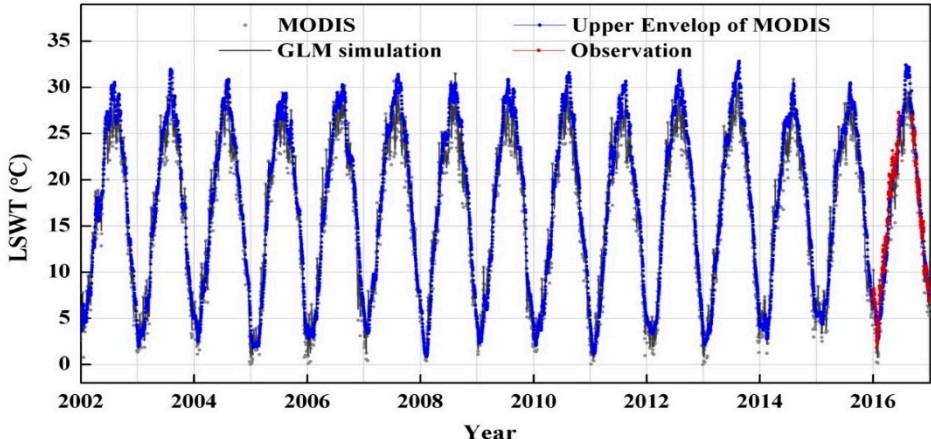

**Figure 8.** Comparison of daily LSWT between MODIS, MODIS UE and GLM simulation from 2002 to 2016 in Lake Chaohu. The grey symbols are the MODIS-derived LSWT; the blue line is the MODIS UE (Upper Envelop line)-derived LSWT, and the black line is the GLM simulation. In situ records shown as the red line are also added into the comparison.

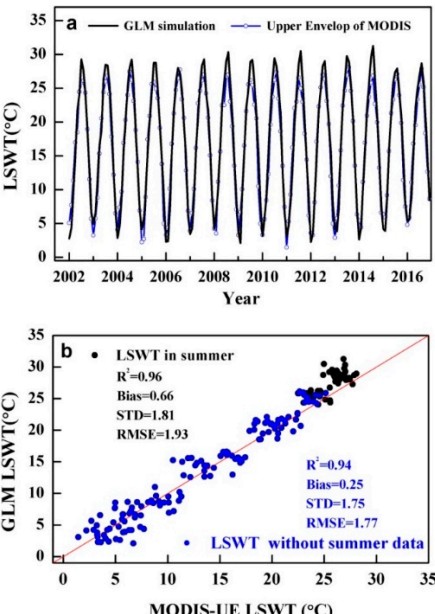

**Figure 9.** Comparison of monthly LSWT between GLM simulation and MODIS LSWT from 2002 to 2016. Panel (**a**): The plot of monthly time-series of MODIS–UE LSWT (blue symbols) and GLM simulation (black line). Panel (**b**): the scatterplot of comparison between monthly GLM simulation and monthly MODIS–UE LSWT. The blue dots represent the comparison of MODIS–UE temperature and GLM simulation without the summertime data (accordingly the statistical information given in blue), while the black dots represent the comparison in summertime data from 2002 to 2016 (accordingly the statistical information given in black).

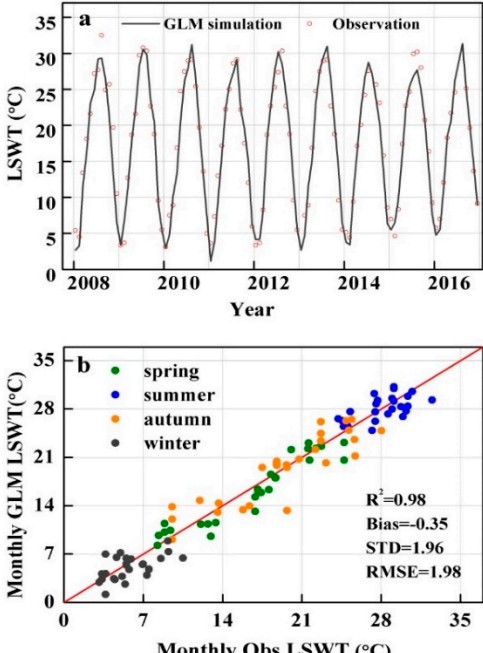

**Figure 10.** Comparison of monthly LSWT between GLM simulation and in situ measurement from 2008 to 2016. Panel (**a**): The plot of the monthly time-series of in situ records (red symbols) and the GLM simulation (black line). Panel (**b**): the scatterplot of the comparison between GLM simulation and monthly observation LSWTs. The green dots represent the monthly value in spring (March, April, and May). The blue dots represent summer (June, July, and August). The orange dots represent autumn (September, October, and December). The black dots represent winter (December, January, and February).

To validate the performance of GLM simulations during summer, the monthly observed LSWT from 2008 to 2016 were also used to assess the model results (Figure 10). The seasonal pattern of simulations was consistent with the observed LSWT. The statistical result showed that the model could reflect the variation of temperatures well during the summer, with a bias of −0.03 °C and an STD of 1.82 °C (Table 2). The performance of LSWT simulation looked more convincing in spring and winter than in autumn. The RMSE value was low (1.73 °C) in spring.

**Table 2.** Statistic information about the seasonal comparison between general lake model (GLM) simulation and observed LSWT from 2008 to 2016. The bias, standard error (STD), and root mean square error (RMSE) were calculated.

| Season | Bias | STD | RMSE |
|--------|------|-----|------|
| Spring | −0.56 | 1.66 | 1.73 |
| Summer | −0.03 | 1.82 | 1.78 |
| Autumn | −0.01 | 2.46 | 2.42 |
| Winter | −0.79 | 1.79 | 1.92 |

The annual anomaly values of MODIS–UE LSWT data and GLM simulations from 2002 to 2016 showed a good agreement (Figure 11), with a correlation coefficient of 0.88, a bias of 0.75 °C, and a standard deviation of 0.13 °C. The annual anomaly observations from 2008 to 2015 were also compared with MODIS–UE LSWT and simulations. The STD value was 0.13 °C compared to MODIS data and 0.34 °C compared to observation. In general, these comparisons showed that the GLM model could provide LSWT reliably and accurately. Thus, GLM could be used for long-term simulations to detect changes in LSWT.

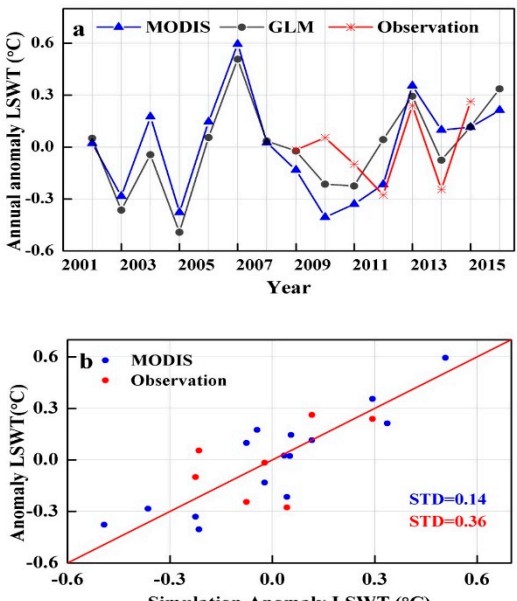

**Figure 11.** Comparison of annual anomaly LSWT between GLM simulation and MODIS LSWT. Panel (**a**): the plot of time series of annual anomaly LSWT from the GLM simulation (black line) and MODIS (red line) from 2002 to 2016, and the observation from 2008 to 2015. Panel (**b**): the scatterplot of the comparison of LSWT from simulation vs. MODIS and simulations vs observations.

### 4.3. Long-Term Trends of LSWT and Atmospheric Forcing

With calibrated parameters, GLM was forced by meteorological variables from 1960 to 2016 to simulate LSWT in Lake Chaohu. As a result, annual LSWT showed a slightly increasing trend of 0.005 °C/year over the whole period. A change point occurred in 1980 (see Figure 12). There was a clear decreasing trend of 0.08 °C/year before 1980, and a significant increasing trend of about 0.05 °C/year after 1980.

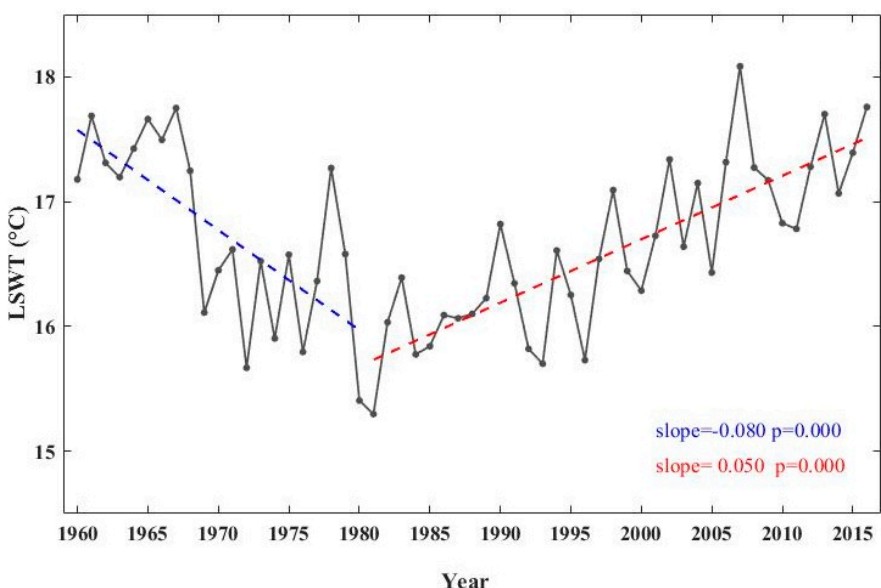

**Figure 12.** Annual LSWT during the period of 1960–2016 from the GLM simulation. Linear trends are given over the period of 1960–1980 (blue dashed line) and of 1981–2016 (red dashed line).

The decadal variations of air temperature and wind speed showed a close relationship with the trend of simulated LSWT. The Pearson coefficient was high, with a value of 0.60 (air temperature),

−0.73 (wind speed), respectively. Before 1980, solar radiation dropped sharply by 1.32 W/m²/year and the wind speed increased by 0.02 m/s/year (see Figure 2). Air temperature and longwave radiation showed a decreasing trend. These changes could have a cooling effect on the lake surface and would lead to a decreased LSWT in Lake Chaohu. Over the period of 1981–2016, a rapid increase of air temperature (0.046 °C/year), a decrease of wind speed (0.029 m/s/year), a sharp increase of longwave radiation (0.288 W/m²/year) and an insignificant decrease of shortwave radiation (−0.203 W/m²/year) became visible. The increasing energy from radiation contributed to heating the lake surface and the combination of decreasing wind speed and increasing air temperatures enhanced the stability of lake stratification.

To further quantify the effect of meteorological variables, which were mainly responsible for the simulated LSWT trends, we detrended the forcing variables by multi-annual mean value. Simulations were performed for every single detrended variable while all other input variables were retained. Three forcing variables were selected: air temperature, wind speed, and shortwave radiation. LSWT simulated by detrending shortwave radiation showed a significant positive trend of 0.14 °C/10year (Figure 13). The LSWT driven by detrending air temperature and wind speed produced a negative trend of −0.12 °C/10year and −0.10 °C/10year, respectively. The simulations for any other detrended input, such as relative humidity, did not show any significant difference to the original simulation. Furthermore, one simulation was performed where wind speed, air temperature, and shortwave radiation were detrended together, which resulted in an insignificant decrease of −0.01 °C/10year of LSWT (Figure 13d).

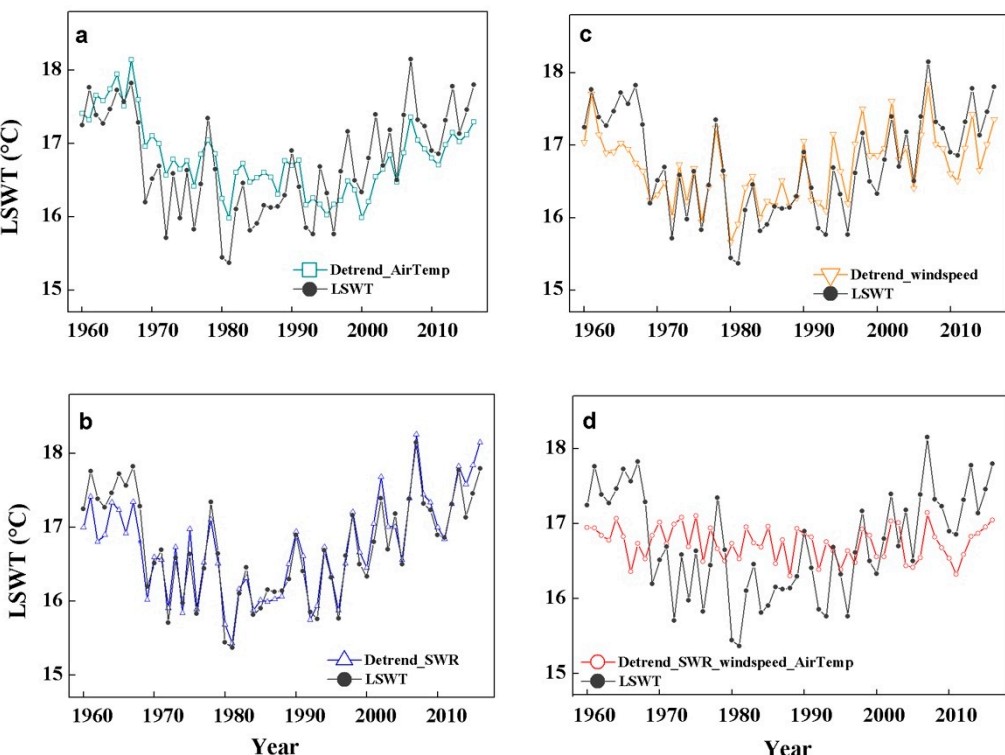

**Figure 13.** The plot of variation of simulation LSWT compared with detrending analysis from 1960 to 2016: the simulated LSWT forced by non-detrend meteorological input (black dot line) and forced by detrending air temperature (**a**), shortwave radiation (**b**), wind speed (**c**), and forced by detrending jointly variables (air temperature, wind speed, shortwave radiation) (**d**).

## 5. Discussion

Compared with our ground truthing of in situ LSWT, the MODIS data revealed a noticeable cool bias at the six sample sites, which also had appeared in many studies retrieving LSWT from MODIS products.

Values ranged from 0.1 °C to several Celsius degrees, such as Lake Taihu (−0.63 °C~−1.53 °C) [87], Lake Urmia (−0.27 °C) [54], the Great Salt Lake (−1.5 °C) [88], Lake Tahoe (−0.1 °C) [35], Lake Vänern and Lake Vättern (~−0.4 °C) [89]. One of the possible factors for the cool bias was skin-bulk temperature difference. Skin temperature was theoretically cooler than the bulk skin [13], because warming from solar irradiation penetrated deeper (in the order of a meter) into the water column and the heat, e.g., from evaporation, was instantly only taken from a very thin layer at the surface. Our data show a smaller cool skin effect during daytime than nighttime (Table 1). We argued the bulk layer became warmer than the skin layer as solar heating penetrated instantaneously deeper than surface cooling. Apparently, skin-water temperature effect on MODIS was stronger in summer, possibly because of the peak value of shortwave radiation and lower wind speed in summer [54,88]. It is possible that poor atmosphere and undetected cloud pixels could also contribute because cloud top temperature would be mistaken as LSWT value, but it was much colder than LSWT [29].

The cool bias from satellite temperatures had been analyzed over the oceans in previous studies, which were also caused by the heat effect [36,90–92]. They were generally based on physical models and empirical parameterizations [36]. However, the characteristics of lakes were more heterogeneous than oceans when converting skin temperature to bulk temperature. Bias values varied with different conditions, such as the lake size, depth, and environmental conditions. To correct differences of bulk and skin temperature in lakes, the previous literature commonly adopted an estimated or an averaged bias value [87,88], or employed the smoothing filter (Loess or Gaussian filter [26,93], or Empirical orthogonal Function techniques [25]. However, the general smoothing methods were easily affected by abrupt drop values. In our study, the cool bias was corrected by an Upper Envelop smoothing method. This method was firstly used in LAI (leaf area index) research [56] for cool bias correction caused by poor atmospheric conditions (e.g., thick clouds) and undetected cloud pixels. Later it was accepted as an effective approach for reducing the effect of negatively biased noise [57,94]. The upper envelope method worked well as all single measurements were either on the envelope or below it. Hardly any data point was found significantly above the upper envelope. A comparison with in situ measurements revealed most of the temperature dynamics seen in in situ measurements qualitatively and quantitatively, as only some of the data points were affected by the cold bias and the others provided enough support for the envelope. Hence an upper envelope over the MODIS data provided a good solution to fill gaps and reduce uncertainty.

Calibrated by the upper envelop of MODIS-derived LSWT, the 1-D model GLM simulated LSWTs by well-fit parameters and exhibited a good agreement when validated by MODIS LSWT and observations at the monthly and annual scale. Especially, the simulations also showed a good performance in summer compared with monthly observations when MODIS under-estimated LSWT. The GLM model could reflect the variability of LSWT in Lake Chaohu accurately and could hindcast the long-term trend of LSWT with the long record of meteorological input data: the simulated LSWT initially decreased at a rate of 0.8 °C/10year from 1960 to 1980, and then increased 0.5 °C/10year after 1980 in Lake Chaohu. The rate of increasing LSWT in the latter phase was also consistent with but higher than the global lake warming trend (0.45 °C/10 year) from the observed and satellite records [10,12,95]. This strong increase of LSWT after 1980 had previously been found in shallow lakes in other regions, such as in northeastern United States [96,97], and northern Europa [61,98].

We detected a breakpoint in the LSWT simulation of Lake Chaohu in 1980, which has also been found in European lakes [61,99]. This shift corresponded to a climate regime shift in the 1980s [59], under the background of rapid warming [100], brightening and dimming [101], a decline in wind speed [102] at a global scale. In Lake Chaohu, air temperature increased by 0.46 °C/decade since 1980, at the rate twice of the global average (0.25 °C/decade) [100], and the wind speed decreased by 0.29m/s/decade, consistent with the atmospheric still phenomenon around the world [102]. Solar radiation decreased, under the background of a decrease in China (1.06 W/m$^2$/decade) [70], contrary to the global brightening, especially in Europa [101]. Previous studies had discussed that higher air temperatures significantly contributed to recent warming of lakes [103,104]. In addition, the change

in solar radiation [17,98], and wind speed [105,106] affected temperature trends in lakes. Our model results suggested that wind speed, air temperature, and solar radiation had an associated effect on the shift and the long-term trend of LSWT in Lake Chaohu, which helped us to understand the importance of meteorological drivers.

It had been reported that Lake Chaohu had suffered from eutrophication and serious cyanobacteria bloom after the 1980s [107,108], coinciding with the breakpoint of LSWT in 1980s. Apparently, the increased temperature played an important role [38,109,110]. The higher temperature (above 25 °C/) generally had led to increased cyanobacteria growth rates and reduced vertical mixing.

## 6. Conclusions

In this study, remote-sensing data (MODIS temperature products) and a well-calibrated lake model named GLM were jointly implemented to investigate the variability of LSWT in Lake Chaohu, a large and shallow lake in China. For ground truthing, in situ data from six locations in the lake were used to assess MODIS LSWT performance by real-time comparisons. According to the comparison, MODIS-derived temperature reflected the variation of lake surface temperature well, with a correlation coefficient of 0.96. However, we also detected a cool bias of 1.25 °C/ when validated by in situ records. As a consequence, MODIS-derived LSWTs were modified by a three-point "Upper Envelop" smoothing method, and then the time series of MODIS-derived LSWT with the bias correction was used as a data source to evaluate the lake model performance. A well-calibrated GLM was used to hindcast a long-term record of LSWT back to 1960 using meteorological data from the weather station. From 1960 to 2016, LSWTs decreased by 0.08 °C/year from 1960 to 1980 and then increased by 0.05 °C/year since 1980, mainly due to a combined effect of rapidly increased air temperature (0.05 °C/year), and decreased wind speed (0.03 m/s/year), and lower shortwave radiation ($-0.22$ W/m$^2$/year).

In conclusion, one-year records of detailed LSWT measurements were sufficient to calibrate the lake model GLM with meteorological data from a nearby weather station. The simulation of LSWT could be extended back as far as the meteorological data were available. MODIS LSWT data was suited for validating the model back to 2002. In summer in particular, deviations could be attributed to MODIS cold bias. Numerical simulations could bridge data gaps in satellite observations and not be affected by atmospheric conditions which influenced both LWST and data collection in the case of satellite measurements. A similar approach may be implemented at other lakes of regional importance. Further work is needed to focus on exploring the links between a warming LSWT and algal concentration in such large and eutrophic lakes under climate change.

**Author Contributions:** X.Z., B.B., and K.W. conceived the study. X.Z. conducted the analysis and wrote the manuscript. M.A.F. contributed to lake observed data collection. All authors discussed the results, commented on the manuscript, and reviewed the final manuscript. All authors have read and agreed to the published version of the manuscript.

**Funding:** This study was funded by the National Key Research and Development Program of China (2017YFA0603601), the National Natural Science Foundation of China (41525018 and 41930970), and the Fundamental Research Funds for the Central Universities (312231103). The latest observational data were obtained from the CMA (http://www.cma.gov.cm). The field data were acquired in the project "Managing Water Resources for Urban Catchments" (grant number 02WCL1337A). This work is also supported by scholarship from China Scholarship Council (No.201706040196).

**Acknowledgments:** The authors would like to thank the data sharing from Kong Xiangzhen. We greatly appreciate the Editor and the anonymous reviewers for their valuable comments.

**Conflicts of Interest:** The authors declare no conflict of interest.

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
