# Peer review of "Reconstructing Six Decades of Surface Temperatures at a Shallow Lake"

_water, doi:10.3390/w12020405_

Round 1

Reviewer 1 Report

The topic sounds interesting and attractive to readers. However, the manuscript needs a significant improvement. Please address the following comments:

Line 31: Define a in  “℃/a”.

Line 61: Grammar error in “to investigated”, please revise.

Introduction: Try to include a discussion of advantages, disadvantages, and limitations of the use of remote sensing.

Introduction: I suggest adding some recent articles published in “Water” that addressed integration of remote sensing and numerical modeling.

Introduction: More description should be discussed about the study area such surface area, location, etc.

Section 2.1: Please define all six inflows that feed the lake, if applicable.

Section 2.2: Include more info about data duration used in the study.

Line 160: Using a window-pixels of (3*3) is a good approach, please add a reference.

Define parameters in equation 1.

Section 3.2: Refer to figure 12 for visual representations of input data.

Line 268-269: Justification should be supported with a reference.

Figure 4-a: Discuss the gap in observation data during the summer time.

Figure 6-a: GLM simulation showed a peak difference compared to the Upper Envelop of MODIS. Calibration of some sensitive factors could significantly enhance the difference.

Figure 8-b: Add a legend to distinguish between blue and black dots.

Line 431: Grammar correction is needed.

Line 431 and 440: formatting is needed, “Bold” formatting is observed.

Line 480 “Conclusion”: Grammar corrections are needed. Please revise.

Author Response

Reviewer #1

Comments and Suggestions for Authors: The topic sounds interesting and attractive to readers. However, the manuscript needs a significant improvement. Please address the following comments:

Line 31: Define a in “°C/a”.    

Response: It has been changed as “°C/yr”.

Line 61: Grammar error in “to investigated”, please revise.

Response: It has been revised as “to investigate

Introduction: Try to include a discussion of advantages, disadvantages, and limitations of the use of remote sensing; I suggest adding some recent articles published in “Water” that addressed integration of remote sensing and numerical modeling. More description should be discussed about the study area such surface area, location, etc.

Response: Thanks for the reviewer comments. In the revised version, we included additional papers from WATER in our citations (References [14, 41, 50, and 81]). Advantages, disadvantage and limitations of remote sensing were introduced in line 56-68: “Remote sensing appears to be an attractive way to complement traditional measurement with a fine spatio-temporal resolution. Several regional-scale LSWT products are also available with the acceptable accuracy, e.g. LSWT for European Alpine lakes (1989-2013) using the Advanced Very High Resolution Radiometer (AVHRR) [21, 22]; LSWT for TP lakes using the MODerate Resolution Imaging Spectroradiometer (MODIS) Land Surface Temperature products [23] and AVHRR products [24], Arclake –LSWT products from 1995-2012 using Along Track Scanning Radiometer (ATSR) series [25]. Those satellite products have been widely used to investigate LSWT variation in global and regional scale [16, 26]. Previous studies on MODIS temperature products have demonstrated the reasonable accuracy on land surface [27-29]. It is suggested that the accuracy is even better on water surface due to the large homogeneity of water surface compared to land surface [29].

However, the temporal extension of a LSWT record with satellite data is far from a simple implementation. Firstly, the daily LSWT data from AVHRR or MODIS still suffer from gaps due to the cloud contaminated pixels. For example, the MODIS derived LSWT dataset in Tibetan Plateau lakes is contaminated by the cloud cover, particularly for the nighttime in summer [23]. Several approaches were adopted to reconstruct data over the gaps to build a reliable daily time series of LSWT, such as Harmonic Analysis of time series (HANTS) with fitted values from sinusoidal function [30], Percentile Filter and Lowess Filter [23]. Also, satellite-derived LSWT and in situ measurement represent water temperature of spatially and temporally different scales. While in situ measurements referred to certain depth at one location (referred to bulk temperature), satellite data record the temperature of upper skin layer by averaging over a larger spatial area (referred to skin temperature). The difference between the skin temperature from satellite and the bulk temperature depends on wind speed, atmospheric aerosols, and surface elevation in a complex manner [19, 31-34]. For example, a noticeable difference between the bulk and skin temperature was found in Lake Tahoe [35]. Donlons, et al. [33] found the difference was obvious especially when at low wind speeds and high radiation in the morning. The relationship between skin and bulk temperature was complicated. An enhanced cool bias was observed when skin surface temperature was larger than 28℃ [36]. Given that, it is indeed to make a comparison by utilizing in situ data with high frequency matching the exact acquisition time of the satellite products. It could assess the accuracy MODIS LSWT and make bias correction, and then had a good application of LSWT on the regions where records of observation limited.”

We considered this sufficient for introductory information. Our paper delivers new insights into advantages and disadvantages. Hence the interested reader profits more from the discussion and conclusion section that from additional sentences in our introduction.

Section 2.1: Please define all six inflows that feed the lake, if applicable.

Response: In revised version, we add the information about inflow and outflow briefly “A large portion of the inflow is from Nanfei River, Hangbu River and Yuxi River and one outflow connecting with Yangtze River.” (Lines 126-127). As the inflow and outflow data are not relevant for the surface temperature because their temperature do not differ much from the lake. For this reason, inflows have been neglected in earlier publications of Lake Chaohu surface temperatures and so have we, we explained in revised text “In our simulation, the heat effects of inflows and outflows were neglected, as inflow and outflow temperature differences were less than 1℃ according to previous observations (see Table.1 in Zhang, et al. [80] and Huang, et al. [81]). Also, the previous study showed the in- and outflows in Lake Chaohu had the limited effect on the thermodynamics and currents within the lake [82]. Hence, the effect of in- and outflows on lake surface temperature was be not considered in our simulation, as energy exchange with atmosphere had a more evident impact.” (See Lines 256-262). There is not much point in referring to irrelevant material.

Section 2.2: Include more info about data duration used in the study.

Response: These in-situ temperatures were acquired from 1st January 2016 to 31st December in 2016. The daily average temperature was calculated at each site, and then averaged over all the sites for the entire lake. Unfortunately, the high-frequency data was not available in our study due to a malfunctioning during the summertime (from June to August). Monthly observations were obtained from 2008 to 2016 from Kong, et al. [51], sampled monthly at a depth of 0.2~0.5m near the lake center usually at 10:00 (See Lines 133-140).

Line 160: Using a window-pixels of (3*3) is a good approach, please add a reference.

Response: the reference has been added as “The area of this pixel matrix was large enough to be represent the temperature estimated from MODIS image [26].|”

Define parameters in equation 1.

Response: It has been added:“Where represents MODIS temperature from Upper envelop method, ,and  represent the data on the  th day, the previous time-step and the following time-step.” (Lines 199-201)

Section 3.2: Refer to figure 12 for visual representations of input data.

Response: It has been added.

Line 268-269: Justification should be supported with a reference.

Response: It has been added (Cited the reference [29]).

Figure 4-a: Discuss the gap in observation data during the summer time.

Response: We changed the text to: “The monthly LSWT observation from 2008 to 2016 was also used to validate the performance of MODIS and MODIS-UE LSWT, especially to bridge the data gap during summer (from June to August) due to mal-functioning of the high-frequency loggers.”(See Lines 321-322).

Figure 6-a: GLM simulation showed a peak difference compared to the Upper Envelop of MODIS. Calibration of some sensitive factors could significantly enhance the difference.

Response: Thanks for the reviewer’s comment. We think the difference between MODIS and GLM simulation are possibly due to the lower value of MODIS LSWT during the summertime, as MODIS LSWTs were not fully validated because of the missing observed data in the summer of 2016. Then we employed the monthly observation from 2008 to 2016 and the MODIS LSWT from 2002 to 2016 to validate the performance of GLM simulation, as shown in Figures 7-9. We found the GLM simulation has a good performance, especially in summertime, with a bias of -0.03 ℃, while MODIS has a lower value in summer. The corresponding text can be found in Lines 353-363.

Figure 8-b: Add a legend to distinguish between blue and black dots.

Response: Corrected as suggested and the caption text was changed to “Figure 8. Comparison of monthly LSWT between GLM simulation and MODIS LSWT from 2002 to 2016. Panel (a): The plot of monthly time-series of MODIS-UE LSWT (blue symbols) and GLM simulation (black line). Panel (b): the scatterplot of comparison between monthly GLM simulation and monthly MODIS-UE LSWT. The blue dots represent the comparison of MODIS-UE temperature and GLM simulation without the summer data (accordingly the statistical information given in blue), while the black dots represent the comparison in summer time data from 2002 to 2016 (accordingly the statistical information given in black).”

Line 431: Grammar correction is needed.

Response: Corrected as suggested.

Line 431 and 440: formatting is needed, “Bold” formatting is observed.

Response: Corrected as suggested.

Line 480 “Conclusion”: Grammar corrections are needed. Please revise.

Response: Corrected as suggested.

Reviewer 2 Report

Overall, this is an impressive paper worthy of publication. The authors do an excellent literature review, and not that many studies that synthesize satellite lake temperature data, in situ lake temperature data, and lake models exist.

Despite very good science and writing overall, I still ask for major revisions to address the following points:

1)      Why is a daily time series of lake temperature needed? Please give reasoning for creating daily plots rather than simply using the available observations.

2)      Because this is a shallow lake there will be rapid decreases in surface temperature when the air temperature changes. This should be discussed and the implications for “filling in” daily temperature series using data from other days. How do air temperature and lake temperature variations typically compare to each other? (e.g., seasonal lag)

3)      Please elaborate a bit on your technique for bulk vs skin lake temperature. There is a rich literature on Sea Surface Temperature bulk-skin differences that might want to be discussed briefly. How does your technique for bulk-vs-skin vary from those used on the ocean?

4)      Please explain more detailed why the MODIS and upper envelope are so different and your justification for the upper envelope. Did you discuss doing a seasonally-varying upper envelope (since summer has a larger bias and other seasons less)

5)      A quick edit for awkward sentences should be conducted (‘The’ vs ‘A’; the title should be “Six-decades of surface temperatures…”)

6)      The authors mention that cloud contaminated pixels were found. How were the cloud contamination identified and screened for? Or were they parameterized for through the bias correction? Please clarify – were the clouds even after the Quality Flags were applied and how were the cloud-contaminated pixels identified?

7)      The GLM simulated lake temperature looks to be significantly warmer than the satellite in the summer. The current text discussion of this does not provide adequate explanation of these errors in my opinion, or their potential influence on the decadal trends. Please clarify this section (4.2).

8)      Please clarify the meteorological observations (this is the detrending?) from the model in the GLM trends section.  My understanding is that the variations in the meteorology station along with the fitting parameters were what forced the GLM? Because of the shallow lake, the air temperature weather station observations likely show a lot of the variation over the 6 decades that the lake temperature model is showing, so please clarify if that is the case.

9)      Is the effect of the algae and less mixing in the lake effectively to make the lake behave as if it was a shallower lake when the algae content is high? Please elaborate on how you determined Kw and Secchi Depth over the 60 year simulation and how you varied them over that period, if increasing algae is the primary forcing.

10)   Finally, please compare the lake warming since observed to that from other studies of lake warming across the world. https://agupubs.onlinelibrary.wiley.com/doi/full/10.1002/2015GL066235  Even without any wind or algae effects one would have expected warming over this period, perhaps there are other factors like aerosols, etc decreasing the solar input in this region of the world?

Author Response

Reviewer #2

Overall, this is an impressive paper worthy of publication. The authors do an excellent literature review, and not that many studies that synthesize satellite lake temperature data, in situ lake temperature data, and lake models exist. Despite very good science and writing overall, I still ask for major revisions to address the following points:

1) Why is a daily time series of lake temperature needed? Please give reasoning for creating daily plots rather than simply using the available observations.

Response: The model simulation delivered daily averages; hence LSWT time series from the other two data sources were adjusted to the same time step. We added a related sentence as “We further computed daily LSWT values over the whole area of the lake, which were also used to compare with the lake model simulation at the same time scale.” (See Line 178)

2) Because this is a shallow lake there will be rapid decreases in surface temperature when the air temperature changes. This should be discussed and the implications for “filling in” daily temperature series using data from other days. How do air temperature and lake temperature variations typically compare to each other? (e.g., seasonal lag)   

Response: Thanks for the comments. In our paper, we prefer to focus on a technology to obtain a reliable daily time series of LSWT from MODIS, which can be acted as a data source to calibrate and validate lake model, or to compare with air temperature. To achieve this goal, the outliers, the data gaps and the cool bias in MODIS LSWT product need to be carefully solved, which would be more complicated than only “filling in” the data from other days. In our method, after the outliers of MODIS-derived LSWT detection, we use the linear interpolation method and an Upper Envelop method to minimize the cool bias. The corresponding context can be found “The outliers of MODIS-derived LSWT were examined and rejected when the values were deviated from climatological temperature by more than 3 times the variation.”( Line 182-184), and “our method adopted a simple linear interpolation to fill data gaps firstly, caused by bad-quality pixels, and then adopted the three-point upper envelop smoothing method” (Line 190-199)

As Lake Chaohu is a shallow lake with the mean depth of 3 meter, the lake can be well-mixed easily caused by wind speed. So the lag between lake surface temperature and air temperature would be not a big issue.

3) Please elaborate a bit on your technique for bulk vs skin lake temperature. There is a rich literature on Sea Surface Temperature bulk-skin differences that might want to be discussed briefly. How does your technique for bulk-vs-skin vary from those used on the ocean?

Response: Thanks for the reviewer’s comment. We added the related discussion in Line 477-481:“The cool bias from satellite temperatures have been analyzed over the oceans in previous studies, which also caused by the heat effect [36, 90-92]. They are generally based on the physical models and empirical parameterizations [36]. However, the characteristics of lakes were more heterogeneous than ocean when converting skin temperature to bulk temperature. Bias values varied with different conditions, such as the lake size, depth and environmental conditions.” Hence in our study, we adopted the upper envelop method to correct the cool bias, which minimized the difference between skin temperature and bulk temperature, and also reducing the effect of negatively biased noise due to undetected cloud pixel and poor atmospheric conditions.

4) Please explain more detailed why the MODIS and upper envelope are so different and your justification for the upper envelope. Did you discuss doing a seasonally-varying upper envelope (since summer has a larger bias and other seasons less)

Response: In our study, we firstly interpolated the data gap and corrected the cool bias by an Upper Envelop smoothing method. This method was firstly used in LAI (Leaf Area Index) research [56] for cool bias correction caused by poor atmospheric conditions (e.g., thick clouds) and undetected cloud pixel. Later it was accepted as an effective approach for reducing the effect of negatively biased noise [57, 94]. The upper envelope method worked well as all single measurements were either on the envelope or below it. Hardly any data point was found significantly above the upper envelope. A comparison with in-situ measurements revealed most of the temperature dynamics seen in in-situ measurements qualitatively and quantitatively, as only a part of the data points were affected by the cold bias and the others provided enough support for the envelope. Hence an upper envelope over the MODIS data seemed to provide a good solution to fill gaps and reduce uncertainty. For the suitability of the upper envelop to capture lake surface temperature at most times, we adopted the time-series of upper envelop of MODIS-derived LSWT (referred as MODIS-UE LSWT) as a remotely-sensed reference to compare with modeling results.( See in Lines 486-495)

5) A quick edit for awkward sentences should be conducted (‘The’ vs ‘A’; the title should be “Six-decades of surface temperatures…”)

Response: Thank for the Reviewer’s suggestion. The title has changed as “Reconstructing Six decades Surface Temperatures at a Shallow Lake by Integrating in-situ and Remote Sensing Data into Lake Modeling

6) The authors mention that cloud contaminated pixels were found. How were the cloud contamination identified and screened for? Or were they parameterized for through the bias correction? Please clarify – were the clouds even after the Quality Flags were applied and how were the cloud-contaminated pixels identified?

Response: The revised context clarified the problem about the undetected cloud pixels in Section 2.3 and in Discussion, the corresponding sentences are “Although the cloud-contaminated pixels have been removed by the cloud mask [29], abnormally low values of LSWT were observed owing to the undetected cloud pixels which much colder than lake surface temperature [29].” (Line 186-189).

The effect of cool bias, caused by undetected contaminated pixels and skin-bulk temperature difference and other factors, was reduced by our method of “Upper Envelop”. The performance of this method dealing with cool bias can also be found in Discussion “ In our study, we firstly interpolated the data gap and corrected the cool bias by an Upper Envelop smoothing method. This method was firstly used in LAI (Leaf Area Index) research [56] for cool bias correction caused by poor atmospheric conditions (e.g., thick clouds) and undetected cloud pixel. Later it was accepted as an effective approach for reducing the effect of negatively biased noise [57, 94].” (Line 536-540)

7) The GLM simulated lake temperature looks to be significantly warmer than the satellite in the summer. The current text discussion of this does not provide adequate explanation of these errors in my opinion, or their potential influence on the decadal trends. Please clarify this section (4.2).

Response: We thank the reviewer’s suggestion. There was a difference between GLM simulation and MODIS LSWT in summer of 2016 in Figure 6. MODIS LSWTs were lower than other data, as shown in Figures 7-9: As own observational data are missing for the summer time in 2016, MODIS LSWT cannot be fully validated in summer, and the cool bias is possibly larger due to the high value of shortwave radiation and the low value of wind speed, leading to a large difference between bulk and skin temperature. Hence, to further validate the performance of GLM simulation, the monthly observation from 2008 to 2016 and the MODIS LSWT from 2002 to 2016 were also employed. As shown in Figures 8-9, the GLM simulations were consistent with the monthly observed LSWT, including the summertime. The bias is -0.03℃ during the summer. It is illustrated that the GLM model with fitting parameters can be able to simulate the variation of LSWT, and MODIS had a lower LSWT in summer time in Lake Chaohu. The revised text can be found in Lines 354-379.

8) Please clarify the meteorological observations (this is the detrending?) from the model in the GLM trends section.  My understanding is that the variations in the meteorology station along with the fitting parameters were what forced the GLM? Because of the shallow lake, the air temperature weather station observations likely show a lot of the variation over the 6 decades that the lake temperature model is showing, so please clarify if that is the case.

Response:  the method of the detrended forcing variables is to quantify the effect of meteorological variables on simulated LSWT trends. Firstly, the LSWTs during the period of 1960-2016 were simulated by forcing data from weather station, the results showed in Figure 10. Then, to further quantify meteorological variables which are mainly responsible for the simulated LSWT trend, we detrended the forcing variables by the multi-annual mean value. For example, as shown in Figure 13a, air temperature was detrending and other forcing variables were retained. We concluded that the simulated LSWT trends were most likely caused by a cooling effect of decreased surface incident solar radiation and a warming effect of reduced wind speed.

9) Is the effect of the algae and less mixing in the lake effectively to make the lake behave as if it was a shallower lake when the algae content is high? Please elaborate on how you determined Kw and Secchi Depth over the 60 year simulation and how you varied them over that period, if increasing algae is the primary forcing.

Response: In our simulation, we adopted the range value of Kw for simulation. The range of Kw value was based on the relationship between Secchi Depth and Kw in Lake Chaohu which was investigated in Liu, et al [85]. The Secchi depth (SD) was commonly around 0.3m in spring and 0.48 in late autumn, and then best fit parameters were performed by the statistical criteria Bias, RMSE, STD. the detail can be found in Lines 263-274 “For most parameter settings, we used default values in our simulation, except for the light extinction coefficient and wind scaling factor. The Secchi depth (SD) was commonly around 0.3m in spring and 0.48 in late autumn [83, 84]. Hence a light extinction coefficient in the range of 3 - 6 m-1 (named “Kw” in GLM) was appropriate, based on the relationship of Kw and Secchi Depth from Liu, et al. [85]. The wind speed records were from the weather station at considerable distance from the lake, and hence a wind scaling factor of 0.6-0.9 was applied based on our knowledge about the uncertainty of parameters and appropriate limits from previous application of GLM ([65], Weber, et al. [86]. In our study, the best fit parameters were calibrated and adjusted to minimize the RMSE based on comparison of simulated data with the observed data in 2016. The observed data refer to both daily time series of MODIS derived LSWT and in situ data. The validation and evaluation of GLM model with best fit parameters were performed from 2002 to 2016 by the statistical criteria RMSE, Bias, STD.”

10) Finally, please compare the lake warming since observed to that from other studies of lake warming across the world. https://agupubs.onlinelibrary.wiley.com/doi/full/10.1002/2015GL066235   Even without any wind or algae effects one would have expected warming over this period, perhaps there are other factors like aerosols, etc decreasing the solar input in this region of the world?

Response: Thanks for the comment. The comparison was written in Lines 505-509, “The rate of increasing LSWT in the latter phase was also consistent with but higher than the global lake warming trend (0.45℃/10yr) from the observed and satellite records [10, 12, 95]. This strong increase of LSWT after 1980 had previously been found in shallow lakes in other regions, such as in northeastern United States [96, 97], and northern Europa [61, 98].”, including the reference the reviewer suggested.

The other factors that may be related to the warming of the lakes also discussed, including the shortwave radiation, wind speed and ecological factors, and detailed can be found in Lines 510-527 “We detected a breakpoint in the LSWT simulation of Lake Chaohu in 1980, which had also been found in European lakes [61, 99]. This shift corresponded to a climate regime shift in 1980s [59], under the background of rapid warming [100], brightening and dimming [101], a decline in wind speed [103] at a global scale. In Lake Chaohu, air temperature increased by 0.46℃/decade since 1980, at the rate twice of the global average (0.25℃/decade) [100], and wind speed decreased by 0.29m/s/decade, consistent with the atmospheric still phenomenon around the world [102]. Solar radiation decreased, under the background of a decrease in China (1.06 W/m2/decade) [70], contrary to the global brightening, especially in Europa [101]. Previous studies had discussed that higher air temperatures significantly contributed to recent warming of lakes [103, 104]. In addition, the change in solar radiation [17, 98], and wind speed [105, 106] affected temperature trends in lakes. Our model results suggested that wind speed, air temperature and solar radiation had an associated effect on the shift and the long-term trend of LSWT in Lake Chaohu, which helped understand the importance of meteorological drivers.

It had been reported that Lake Chaohu had suffered from eutrophication and serious cyanobacteria blooms after the 1980s [107 108], coinciding with the breakpoint of LSWT in 1980s. Apparently, the increased temperature played an important role [38, 109, 110]. The higher temperature (above 25℃) generally had led to increased cyanobacteria growth rates and reduced vertical mixing.”

Reviewer 3 Report

The authors are commended for the approach taken in assessing climate change impacts on lake surface temperature. The general approach though needs to be improved to eliminate model uncertainty and to quantify if the assumptions made in the paper are reasonable. These include accounting for inflows, assessing how well the 1D model represents multiple sampling sites on the lake, and reducing model biases. The following comments detail many of these concerns:

Line 53: “to retrieve in conclusion, an approach…” – grammar, reword Line 94: “the dynamics of lake surface temperature can be accurately captured from one-dimensional model.” There are many cases where there are significant longitudinal gradients in lakes and reservoirs. The authors need to show that 1D is not always ideal but can be a default approach usually when field data are lacking. Line 105: “key meteorological factor” – what is the key meteorological factor? Are not there several key factors? Line 127: Why were measurements averaged over a day? Was not the diurnal response important especially when correlating to satellite measurements? What were the differences in the measurements spatially? These are important considerations. Line 220: “Due to the longer availability, the derived shortwave radiation revealed long-term trends” – what long-term trends? Line 230: No error statistics on how well the estimate of short wave solar compared to other data sources. Also, no discussion on what typical cloud cover was estimated to reduce short-wave solar or to augment long-wave radiation. Line 232, 237: There is no mention of inflows or outflows. These must also be significant – how were they estimated and what was their estimated inflow temperature? If they were neglected, how was the water budget and lake water surface computed to agree with field data? This seems like a significant missing part of the modeling study and may invalidate the overall results if not quantified carefully. Line 235: Why use daily average meteorological data when hourly data exist? Figure 5 – There is a significant summer bias of not predicting the warm temperatures. Why is this systematic bias evident and can it be reduced? Section 4.2 and Figure 6 – why the bias also in the summer with the GLM model? The error statistics are not that good. The large bias > 1oC is a serious problem that needs to be improved upon. I assume this is for daily average conditions which should make it easier than hourly to match. The model needs to be improved. Figure 9, Figure 10 – Model comparison to monthly and yearly observation data often obscures weaknesses in model predictive ability. How much confidence can there be in yearly data when there is large bias on observational data? Line 390-393: How can trends be significant if the error of the model is larger than the trend? Lines 399-402: Why were there trends in different meteorological forcing like wind, air temperature, etc.? [OK, I see this in Line 464 – no need to comment on this question.] Also, how reliable are these historical data? There has not been a discussion of the data quality which drives the lake model, nor of the impact of inflows and outflows which have an effect much greater (Line 238) than the trend in the temperature predictions.

Author Response

Reviewer #3

Comments and Suggestions for Authors

The authors are commended for the approach taken in assessing climate change impacts on lake surface temperature. The general approach though needs to be improved to eliminate model uncertainty and to quantify if the assumptions made in the paper are reasonable. These include accounting for inflows, assessing how well the 1D model represents multiple sampling sites on the lake, and reducing model biases. The following comments detail many of these concerns:

Line 53: “to retrieve in conclusion, an approach…” – grammar, reword

Response: It has been revised as “….to retrieve. In conclusion, an approach to extend time series into the past is highly desirable.”

Line 94: “the dynamics of lake surface temperature can be accurately captured from one-dimensional model.” There are many cases where there are significant longitudinal gradients in lakes and reservoirs. The authors need to show that 1D is not always ideal but can be a default approach usually when field data are lacking.

Response: Thanks for the reviewer’s suggestion. In general, vertical temperature effects in shallow lakes (Lake Chaohu) are very small. Dynamical effects, like upwelling of deep water, has a very limited effect on LSWT. We revised the related sentences as “One-dimensional (1-D) physical-based models have been widely used in recent years. Although they cannot provide good solutions about both horizontal and vertical heterogeneity that can be resolved better in rather complex three-dimensional models, 1-D dimensional models can offer a good compromise between computational efficiency and physical reality with minimal calibration requirements and with fewer requirements about input data compared with two- and three-dimensional models [41-45].”(See Lines 92-97)

Line 105: “key meteorological factor” – what is the key meteorological factor? Are not there several key factors?

Response: The sentence has been revised as “Another scientific merit is that the well-calibrated lake model is able to forecast lake surface temperature with reasonable accuracy about the key meteorological factors, such as wind speed and solar radiation.”

Line 127: Why were measurements averaged over a day? Was not the diurnal response important especially when correlating to satellite measurements? What were the differences in the measurements spatially? These are important considerations.

Response: The diurnal LSWT variations were also compared in Section 4.1. The real-time comparison from six sample sites between MODIS products and in situ measurement were shown in Section 4.1, including the daytime and nighttime comparison. Since the GLM model is a 1-D model that the surface temperature was integrated into one value and the output was shown as the daily time step, in situ temperatures were compute into the daily value and also averaged from six sample sites to compare with model simulation from GLM at the same spatial and temporal scale.

Line 220: “Due to the longer availability, the derived shortwave radiation revealed long-term trends” – what long-term trends?

Response: We revised the sentence as “the derived shortwave solar radiation revealed a dimming before 1980s and a brightening afterwards [69, 70, 73].”

Line 230: No error statistics on how well the estimate of short wave solar compared to other data sources. Also, no discussion on what typical cloud cover was estimated to reduce short-wave solar or to augment long-wave radiation.

Response: As the methods to retrieve shortwave radiation and longwave radiation have been described and discussed in detail in previous studies including the error statistics performance and the cloud cover; we give a brief introduction in section 3.2 for the sake of brevity.

Line 232, 237: There is no mention of inflows or outflows. These must also be significant – how were they estimated and what was their estimated inflow temperature? If they were neglected, how was the water budget and lake water surface computed to agree with field data? This seems like a significant missing part of the modeling study and may invalidate the overall results if not quantified carefully.

Response: In revised version, we add the brief information about the inflows and outflows “A large portion of the inflow is from Nanfei River, Hangbu River and Yuxi River and one outflow connecting with Yangtze River” (Line 126-127).

For the lake Chaohu, heating or cooling for solar radiation and wind mixing showed more importance for surface energy balance than the heat dynamics for in and outflows. In- and outflow thus have a less significant effect on lake surface temperature simulation than the lake mixing dynamics simulation. We explain the reason in revised version as “The inflow and outflow temperature difference was less than 1℃ according to the previous observation (see Table.1 in Zhang, et al. [80] and Huang, et al. [81]). Also, the previous study showed the in- and outflows in Lake Chaohu had a limited effect on the thermodynamics and currents in the lake [82]. Hence, the effect of in- and outflows on lake surface temperature would be not considered in our simulation, as energy exchange with atmosphere had a more evident impact than energy exchange.” (Lines 256-262)

Line 235: Why use daily average meteorological data when hourly data exist?

Response: Based on the GLM description, the model with specific parameters ran at the hourly resolution which is recommended by model developer. The daily or hourly meteorological time-series data to force the model is required. In our study, the daily meteorological variables were used for the model which provided by Chinese Meteorological Administration dataset. To be understood clearly, we further revised as “The model started at the first day of simulation period, such as 1st, January of 2016, and then ran forced by daily meteorological time-series data, producing daily water surface temperature

Figure 5 – There is a significant summer bias of not predicting the warm temperatures. Why is this systematic bias evident and can it be reduced?

Response: The reason for the summer bias in Figure 5 is that: Either GLM simulation or MODIS data cannot be validated by daily observed LSWT data because of the loggers were missing. So the performance of GLM simulation was further evaluated by the observed monthly data. the results showed that GLM simulation fit well in summer and MODIS-derived LSWT showed a systematic cool bias (See in Figures 7-9) In our study, we explain this systematic bias in summer in discussion that “Apparently, skin water temperature effect on MODIS was stronger in summer, possibly because the peak value of shortwave radiation and a lower wind speed in summer [53, 88]. Possibly poor atmosphere and undetected cloud pixel could also contribute, because cloud top temperature would be mistaken as LSWT value, but it is much colder than LSWT [29].” (Lines 474-478).

Section 4.2 and Figure 6 – why the bias also in the summer with the GLM model? The error statistics are not that good. The large bias > 1oC is a serious problem that needs to be improved upon. I assume this is for daily average conditions which should make it easier than hourly to match. The model needs to be improved. Figure 9, Figure 10 – Model comparison to monthly and yearly observation data often obscures weaknesses in model predictive ability. How much confidence can there be in yearly data when there is large bias on observational data?

Response: Thank you for the reviewer’s comment. Figure 6 showed the comparison between GLM simulation and MODIS LSWT in 2016. As the missing data in summer time in 2016, MODIS LSWT cannot been fully validated in summer, and the cool bias is possibly larger due to the high value of shortwave radiation and the low value of wind speed, leading to a large difference between bulk and skin temperature. Hence, to further validate the performance of GLM simulation, the monthly observation from 2008 to 2016 and the MODIS LSWT from 2002 to 2016 were also employed. As shown in Figures 8-9, the GLM simulation were consistent with the monthly observed LSWT, including the summertime. The bias is -0.03℃ during the summer. It is illustrated that the GLM model with fitting parameters can simulate the variation of LSWT, and MODIS had a lower LSWT in summer time in Lake Chaohu. The revised context can be found in Lines 354-379 “The variations were similar between MODIS-UE LSWT and GLM simulation. Then LSWT simulation also compared with MODIS-UE LSWT when excluding the data in summertime, with the bias and STD value of 1.09℃ and 1.53℃, respectively (as shown in Figure 6). The large bias of LSWT simulation was possibly contributed to the data in summertime, and a cool bias was found clearly in summertime. As the observation was missing during the summer of 2016, it is difficult to validate whether MODIS-UE LSWTs simulations were correct during the summer, especially the cool bias maybe larger than other season due to the peak value of solar radiation and low wind speed value [53, 88]. Hence, the monthly observation from 2008 to 2016 and the MODIS LSWT from 2002 to 2016 were also employed to validate the GLM simulation performance.

As seen in Figures 7-9, the validation of the GLM simulations were accomplished by comparison with monthly observed LSWT from 2008 to 2016 and with MODIS LSWT from 2002 to 2016. Simulation agreed well with MODIS-UE LSWT (Figure 7). The seasonal cycle was reflected both in MODIS-UE LSWT and simulations, but the LSWT from MODIS showed the large deviation to lower values in summer when compared with GLM simulation. From the data comparison at the monthly scale (Figure 8), the correlation coefficient is 0.96 and the bias amounts to 0.66℃. The correlation coefficient is 0.94 and the bias 0.25℃ when summer values are excluded.

To validate the performance of GLM simulations during summer, the monthly observed LSWT from 2008 to 2016 were also used to assess the model results (Figure 9). The seasonal pattern of simulation was consistent with the observed LSWT. The statistical result showed that the model could reflect the variation of temperatures well during the summer, with a bias of -0.03℃ and an STD of 1.82℃ (Table 2). The performance of LSWT simulation looked more convincing in spring than in autumn, as RMSE value was low (1.73℃) in spring. The winter temperature also showed a good agreement.”

Line 390-393: How can trends be significant if the error of the model is larger than the trend?

Response: Beside the errors from the model itself, the difference of spatial scales among lake model and observation, and the systematic errors from remote sensing could also possibly produce the errors when compared with daily observation. The results in our study showed a good agreement between MODIS and GLM simulation, with a correlation coefficient of 0.88, a bias of 0.75℃, and a standard deviation of 0.13℃, as it is seen in Figure 9 and Figure 10. In general, these comparisons showed that the GLM model could provide LSWT reliably and accurately, and GLM could be used for long-term simulations to detect changes in LSWT. Thus, the trends of simulated LSWT in Lake Chao could be considered as a statistical significance.

Lines 399-402: Why were there trends in different meteorological forcing like wind, air temperature, etc.? [OK, I see this in Line 464 – no need to comment on this question.] Also, how reliable are these historical data? There has not been a discussion of the data quality which drives the lake model, nor of the impact of inflows and outflows which have an effect much greater (Line 238) than the trend in the temperature predictions.

Response: The meteorological data to drive the model is provided from the China Meteorological Administration dataset during the study period of 1960-2016. The dataset was applied to the strict quality control and was of good quality [67]. The thermal effect from inflow and outflow were not included in model simulation, as we explained “In our simulation, the heat effects of inflows and outflows were neglected. The inflow and outflow temperature difference was less than 1℃ according to the previous observation (see Table.1 in Zhang, et al. [80] and Huang, et al. [81]). Also, the previous study showed the in- and outflows in Lake Chaohu had the limited effect on the thermodynamics and currents within the lake [82]. Hence, the effect of in- and outflows on lake surface temperature would be not considered in our simulation, as energy exchange with atmosphere had a more evident impact than energy exchange.”(See the Lines 256-262)

Round 2

Reviewer 1 Report

The revised manuscript showed a significant improvement. However, extra eyes are needed for grammar and formatting issues as noticed in many occasions (example: lines 1074 through 1084) and many others throughout the text; especially in the new red texts and in conclusions.

Also, figure 12 has not been referred in section 3.2 as recommended in the previous review.

Author Response

Reviewer#1

Comment: The revised manuscript showed a significant improvement. However, extra eyes are needed for grammar and formatting issues as noticed in many occasions (example: lines 1074 through 1084) and many others throughout the text; especially in the new red texts and in conclusions.

Also, figure 12 has not been referred in section 3.2 as recommended in the previous review.

Response: We appreciate the comments from the reviewer. The grammar and format have been carefully checked. And the Figure 12 has been referred to in this revised version “The climatological information of six variables can also be found in Figure 12.” (Lines 203-204)

Reviewer 2 Report

The authors did a good job responding and I only suggest 2 minor changes:

1. Did you discuss doing a seasonally-varying upper envelope? (add a few sentences about why this was not done or as a possible future work.

2. For paper title, I think:

Reconstructing Six-decades OF Surface Temperature at a shallow lake" is best.

Author Response

Reviewer#2:

Comments: 1. Did you discuss doing a seasonally-varying upper envelope? (add a few sentences about why this was not done or as a possible future work.

Response: The seasonal performance of upper envelop curve has been added as “The bias is -2.05℃ in summer, higher than other spring (-1.09℃.), autumn (-0.77℃) and winter (-0.55℃).” (see Lines 295-296)

For paper title, I think: Reconstructing Six-decades OF Surface Temperature at a shallow lake" is best.

Response: Thank for the suggestion. The title has been changed

Reviewer 3 Report

Review of “Reconstructing Six-decade Surface Temperatures at a Shallow Lake…” by Zhang et al.

This is the re-review of this paper. This is in response to the original comments made by the authors. The following comments detail my continued items that need clarification. It is clear that I have issues with neglecting inflows and averaging model results over long periods of time since I think it reduces the usefulness of the model. Many others – like in this study – have found that long term averages reduces bias and error which should apply to most model studies. Even though I have those biases, I suggest that the paper be published with a few comments on how to improve the overall modeling technique for a future study. Here are a few final comments:

Old Line 105: “key meteorological factor” – what is the key meteorological factor? Are not there several key factors? - - Resolved but need to revise grammar in new sentence. “…with reasonable accuracy from using the key meteorological variables factors, such as wind speed and solar radiation.” Old Line 127: Why were measurements averaged over a day? Was not the diurnal response important especially when correlating to satellite measurements? What were the differences in the measurements spatially? These are important considerations. – Not resolved. The authors did not show what the variability in measurements longitudinally was. This is an important metric to know how important small changes in predicted values are or are not. The authors must show what the average difference in temperature was spatially for this to be of value to the reader. Old Line 232, 237: There is no mention of inflows or outflows. These must also be significant – how were they estimated and what was their estimated inflow temperature? If they were neglected, how was the water budget and lake water surface computed to agree with field data? This seems like a significant missing part of the modeling study and may invalidate the overall results if not quantified carefully. – Not completely resolved. Since this has an effect of 1oC which is large relative to your changes in lake temperatures, the authors need to state in the conclusions that a way to improve this study is to account for inflows and outflows which may affect the results of this study.

Author Response

Reviewer#3

Comment: Old Line 105: “key meteorological factor” – what is the key meteorological factor? Are not there several key factors? - - Resolved but need to revise grammar in new sentence. “…with reasonable accuracy from using the key meteorological variables factors, such as wind speed and solar radiation.”

Response: as reviewer’s suggestion, the sentence has been change to “Another scientific merit is that the well-calibrated lake model can forecast lake surface temperature with reasonable accuracy using key meteorological variables, such as wind speed and solar radiation.” (Lines 96-98).

Old Line 127: Why were measurements averaged over a day? Was not the diurnal response important especially when correlating to satellite measurements? What were the differences in the measurements spatially? These are important considerations. – Not resolved. The authors did not show what the variability in measurements longitudinally was. This is an important metric to know how important small changes in predicted values are or are not. The authors must show what the average difference in temperature was spatially for this to be of value to the reader.

Response: In revised version, we added the related sentences in section 4.1 as “Based on the observation in six sample sites, the highest average temperature was 17.34℃ and the lowest value was 16.92℃. Also, the bias varied from -0.50℃ to -0.99℃ at daytime and from -1.22℃ to -2.12℃ at nighttime between observation and MODIS data among six sample sites. Hence, the heterogeneous temperature was small on the surface layer.. (see Lines 279-282)

Old Line 232, 237: There is no mention of inflows or outflows. These must also be significant – how were they estimated and what was their estimated inflow temperature? If they were neglected, how was the water budget and lake water surface computed to agree with field data? This seems like a significant missing part of the modeling study and may invalidate the overall results if not quantified carefully. – Not completely resolved. Since this has an effect of 1oC which is large relative to your changes in lake temperatures, the authors need to state in the conclusions that a way to improve this study is to account for inflows and outflows which may affect the results of this study.

Response: Our consideration about the problem of the effect of different inflow temperatures is that: the residence times in Lake Chaohu are around 180 days [50]. If we assume inflow water takes 1 day to adjust temperature to lake conditions and we mix ALL inflowing water into the Western basin at MAXIMUM temperature difference of 1°C, the effect of temperature should be estimated by 1/(180/2)*1°C, should be less than 0.01°C at maximum. We included the residence time in the text to supply a quantification. Lines 226-232 read now: “In our simulation, the heat effects of inflows and outflows were neglected, as inflow and outflow temperature differences were less than 1℃ according to previous observations (see Table.1 in Zhang, et al. [80] and Huang, et al. [81]) ,and residence times (~180 days [50]) were too long for a visible effect on water temperature. Also, the previous study showed the inflows and outflows in Lake Chaohu had a limited effect on the thermodynamics and currents in the lake, while energy exchange with atmosphere on lake surface had a more evident impact [81]. Hence, the effect of in- and outflows on lake surface temperature was not considered in our simulation.”